# The Analytical Objective Hysteresis Model (AnOHM v1.0): Methodology to Determine Bulk Storage Heat Flux Coefficients

Ting Sun[1,2,3*], Zhi-Hua Wang[4], Walter C. Oechel[5,6], Sue Grimmond[1*]

[1]Department of Meteorology, University of Reading, Reading RG6 6BB, UK
[2]Department of Hydraulic Engineering, Tsinghua University, Beijing 100084, China
[3]State Key Laboratory of Hydro-Science and Engineering, Tsinghua University, Beijing 100084, China
[4]School of Sustainable Engineering and the Built Environment, Arizona State University, Tempe, AZ 85287, US
[5]Global Change Research Group, Department of Biology, San Diego State University, San Diego, CA 92182, US
[6]Department of Environment, Earth and Ecosystems, The Open University, Milton Keynes MK7 6AA, UK

*Correspondence to*: Ting Sun (ting.sun@reading.ac.uk); Sue Grimmond (c.s.grimmond@reading.ac.uk)

**Abstract.** The net storage heat flux is not only a large part of the urban surface energy balance (SEB) but its determination remains a significant challenge. The diurnal hysteresis behaviour found between the net storage heat flux ($\Delta Q_S$) and net all-wave radiation ($Q^*$) has been captured in the Objective Hysteresis Model (OHM) parametrization of $\Delta Q_S$. Although, successfully used in urban areas, the limited availability of coefficients for OHM hampers its application. To facilitate use, and enhance physical interpretations of the OHM coefficients, an analytical solution of the one-dimensional advection-diffusion equation of coupled heat and liquid water transport in conjunction with the SEB is conducted, allowing development of AnOHM (Analytical Objective Hysteresis Model). A sensitivity test of AnOHM to surface properties and hydrometeorological forcing is presented using a stochastic approach (the Subset Simulation). The sensitivity test suggests that albedo, Bowen ratio and bulk transfer coefficient, solar radiation and wind speed are most critical. AnOHM, driven by local meteorological conditions at five areas with different land use, is shown to simulate the $\Delta Q_S$ flux well (RMSE values of ~30 W m$^{-2}$). The intra-annual dynamics of OHM coefficients are explored. AnOHM offers significant potential to enhance modelling of the surface energy balance over a wider range of conditions.

## 1 Introduction

The essential role of an integrated land surface model is to physically predict the land-atmosphere interactions by resolving the transfer of energy, water, and trace gases (Katul et al., 2012; Liang et al., 1994; Sellers et al., 1997). Such land-atmospheric interactions are strongly modulated by the solar energy partitioning at the land surface (Chen and Dudhia, 2001; McCumber and Pielke, 1981; Yang and Wang, 2014). The surface energy balance (SEB) equation is (Oke, 1988):

$$Q^* - \Delta Q_S = Q_H + Q_E \tag{1}$$

where $Q^*$, $\Delta Q_S$, $Q_H$ and $Q_E$ are the net all-wave radiation, net storage, turbulent sensible and latent heat fluxes, respectively. Eqn (1) distinguishes the available energy at the land surface (left hand side) from the heat transfer through turbulent transport (right hand side).

The turbulent and radiative fluxes are more readily measured using standard techniques (e.g., eddy-covariance instruments, radiometry, etc.) than $\Delta Q_S$ because of spatial sampling associated with the instrumental techniques (Offerle et al., 2005; Pauwels and Daly, 2016; Roberts et al., 2006; Wang, 2012). This is because for $\Delta Q_S$ the net energy stored or released by changes in sensible heat within the canopy air layer, roughness elements (RE, e.g. vegetation, buildings) and the ground have to be considered. The volume of interest extends from the top of the roughness sub-layer to the depth in the ground where the

daily averaged vertical net heat conduction is zero (see Figure 2 in Masson et al., 2002).

Knowledge of $\Delta Q_S$ is crucial to a wide range of processes and applications: from modelling turbulent heat transfer and boundary layer development to predicting soil thermal fields, *etc.*. In rural or simple bare soil sites, this term may be a small fraction of the net all wave radiation (Oliphant et al., 2004) but in areas where there is more mass, such as cities, the term

becomes much more significant (Kotthaus and Grimmond, 2014a). In urban areas, the large amount of mass made from high heat-admitting materials is arranged in canyon morphology. These features are critical in causing the urban heat island, through radiative trapping and the thermal inertia due to this storage term.

A wide range of techniques, however, are available to obtain $\Delta Q_S$ in urban systems (Grimmond et al., 1991; Roberts et al.,

2006). These include:

a) Heat conduction approach: the weighted average of heat flows through all urban materials and surfaces by solving heat conduction equation (e.g., buildings, streets, vegetated lands, etc.) (Offerle et al., 2005; Wang et al., 2012; Yang et al., 2014);

b) Thermal mass scheme: the storage heat is inferred from the changes in thermal mass of all components of the urban

system (Kerschgens and Kraus, 1990).

c) Composite of heat flux plates: Kerschgens and Hacker (1985) and Kerschgens and Draushke (1986) combined measurements from grass and paved surfaces;

d) Parameterisation of $Q^*$: linear function of $Q^*$ (Oke et al., 1981); hyperbolic (*cotangent*, *secant*) function (Doll et al., 1985); and hysteresis relation between $\Delta Q_S$ and $Q^*$ (Camuffo and Bernardi, 1982) used in the Objective Hysteresis

Model (OHM) (Grimmond et al., 1991)

Practical difficulties of direct measurement of $\Delta Q_S$ in urban areas, result in the SEB residual (i.e., $Q^* + Q_F - (Q_H + Q_E)$) frequently being the "best" observations (Ching et al., 1983; Doll et al., 1985; Oke and Cleugh, 1987)(Ao et al., 2016; Li et al., 2015), where $Q_F$ is the anthropogenic heat flux.

The focus here is on the OHM approach, which is forced by $Q^*$ and accounts for the diversity of the surface by 2 or 3-dimensional weighting ($f$) of the sub-facets ($i$) materials (Grimmond et al., 1991):

$$\Delta Q_S = \sum_i f_i \left( a_{1,i} Q_i^* + a_{2,i} \frac{\partial Q_i^*}{\partial t} + a_{3,i} \right) \tag{2}$$

where the $a_1$, $a_2$ and $a_3$ coefficients are for individual facets determined by least square regression between $\Delta Q_S$ and $Q^*$ using results from observation (e.g. asphalt road (Anandakumar, 1999), wetland (Souch et al., 1998), forests (Oliphant et al., 2004)) or numerical modelling (e.g. urban canyons (Arnfield and Grimmond, 1998), roofs (Meyn and Oke, 2009)). The coefficients provide a net behaviour of a facet type in a typical setting, rather than being required to identify the component materials within a facet (e.g. multiple materials making up a roof, wall, with varying thermal connectivity and individual properties). As such, OHM features the less demanding parameterisations and more direct understanding of control of $\Delta Q_S$ by $Q^*$ compared with other approaches. Despite the shortage of OHM coefficients for the wide range facet types, OHM captures the urban $\Delta Q_S$ overall generally well (Grimmond and Oke, 1999; Järvi et al., 2011; 2014; Karsisto et al., 2015; Roth and Oke, 1995).

OHM is a cornerstone in the urban land surface models, Surface Urban Energy and Water Balance Scheme (SUEWS) (Järvi et al., 2011; 2014; Ward et al., 2016) and Local-scale Urban Meteorological Parameterisation Scheme (LUMPS) (Grimmond and Oke, 2002), and plays an essential role in determining the initial energy partitioning at each time step of the models' simulations. Previous modelling studies (Arnfield and Grimmond, 1998; Meyn and Oke, 2009) have led to better understanding of the OHM coefficients. Solution of the one-dimensional advection-diffusion equation of coupled heat and liquid water transport by Gao et al. (2003; 2008) was used to explore the physical relation of OHM coefficients $a_1$ and $a_2$ to the phase lag between $\Delta Q_S$ and $Q^*$. However, insight into $a_3$ remains unclear (Sun et al., 2013).

In this paper, the solutions of the one-dimensional advection-diffusion equation of coupled heat and liquid water transport (Gao et al., 2003; 2008) are employed with the SEB (eqn 1) to investigate more fully the three OHM coefficients, the outcomes of which lead to development of the Analytical Objective Hysteresis Model (AnOHM) (section 2). Then the Monte Carlo-based Subset Simulation (Au and Beck, 2001) approach is used to undertake a sensitivity analysis of AnOHM to surface properties and hydrometeorological conditions (section 3). An offline evaluation of AnOHM's performance for five sites with different land covers (section 4) allows us to conclude that this is an alternative approach to obtain OHM coefficients. Given this allows applications across a much wider range of environments and meteorological conditions, AnOHM has important implications for land surface modelling (urban and non-urban).

## 2 Model Development

### 2.1 Parameterization of Storage Heat Flux $\Delta Q_S$ for a Land Surface

For a given land surface (e.g. bare soil), the governing heat conduction-advection equation can be written (Gao et al., 2003; 2010):

$$\frac{\partial T}{\partial t} = \lambda \frac{\partial^2 T}{\partial z^2} + W \frac{\partial T}{\partial z} \tag{3}$$

where $T$ is the temperature at a reference depth $z$ (positive downward), $t$ is time, $\lambda$ is the thermal diffusivity and $W = \partial\lambda/\partial z - (C_W/C_g)w\varphi$ is the soil water flux density (Ren et al., 2000), with $C_W$ the volumetric heat capacity of water, $C_g$ the volumetric heat capacity of soil, $w$ the pore water velocity, and $\varphi$ the volumetric soil water content.

The steady-periodic solution of equation (3) corresponding to the principle Earth's rotation frequency ($\omega = 2\pi/24$, in

rad h$^{-1}$), with boundary condition:

$$T_S = A_{T_S} \sin(\omega t - \gamma) + \overline{T_S}, \tag{4}$$

is given by:

$$T(z,t) = A_{T_S} \exp(-z/M) \sin(\omega t - z/N - \gamma) + \overline{T_S} \tag{5}$$

where $M = \frac{2\lambda}{\Delta+W}$, $N = \frac{\Delta}{\omega}$ and $\Delta = \sqrt{\frac{W^2+\sqrt{W^4+16\lambda^2\omega^2}}{2}}$; and $\overline{T_S}$, $A_{T_S}$ and $\gamma$ denote the daily mean value, amplitude and initial phase of surface temperature, respectively, which need to be determined by the boundary conditions imposed by the SEB.

From Fourier's law, the soil heat flux is then given by:

$$G(z,t) \equiv -\frac{k\partial T}{\partial z} = kA_{T_S} \frac{\sqrt{M^2+N^2}}{MN} \exp\left(-\frac{z}{M}\right) \sin\left(\omega t - \frac{z}{N} - \gamma + \delta\right) \tag{6}$$

where $\delta = \arctan\left(\frac{M}{N}\right) = \arctan\left[\frac{2\lambda\omega}{(\Delta+W)\Delta}\right]$ and $k$ is the thermal conductivity. In particular, at the surface $z = 0$, the ground heat flux $G_0$ is given by:

$$G_0(t) = kA_{T_S} \frac{\sqrt{M^2+N^2}}{MN} \sin(\omega t - \gamma + \delta), \tag{7}$$

And a simple written form of $\Delta Q_S$ (if only one surface) can be given as:

$$\Delta Q_S = G_0 = c_\eta \sin(\omega t + \eta) \tag{8}$$

where $\eta = \delta - \gamma$ and $c_\eta = kA_{T_S} \frac{\sqrt{M^2+N^2}}{MN}$.

It is noted although the above derivation only considers the land surface made of a single material type, the derived $\Delta Q_S$ (eqn 8) can be adapted for surfaces made of composite materials or volumes given appropriate bulk/ensemble properties.

## 2.2 Parameterization of Net All-wave Radiation $Q^*$ for a Land Surface

Given the parameterizations of incoming longwave radiation $L_\downarrow$, outgoing longwave radiation $L_\uparrow$, sensible heat flux $Q_H$, latent heat flux $Q_E$, and storage heat flux $\Delta Q_S$ as follows:

$$L_\downarrow = \varepsilon_a \sigma T_a^4, \tag{9}$$

$$L_\uparrow = \varepsilon_s \sigma T_S^4, \tag{10}$$

$$Q_H = C_h U (T_S - T_a), \tag{11}$$

$$Q_E = Q_H / \beta, \tag{12}$$

$$\Delta Q_S = G_0, \tag{13}$$

the boundary condition imposed by the SEB relation can be rewritten as:

$$(1 - \alpha) K_\downarrow + \varepsilon_a \sigma T_a^4 - \varepsilon_s \sigma T_S^4 = C_h U (1 + \beta^{-1})(T_S - T_a) + G_0 \tag{14}$$

where the turbulent fluxes $Q_H$ and $Q_E$ are parameterized as functions of temperature gradient $T_S - T_a$ with albedo $\alpha$, bulk transfer coefficient $C_h$, wind speed $U$ and Bowen ratio ($\beta = Q_H/Q_E$). Theoretically, the reflected part of $L_\downarrow$ (i.e. $(1 - \varepsilon_s)L_\downarrow$)
should be included in formulation of $L_\uparrow$ (i.e., $\varepsilon_s \sigma T_S^4 + (1 - \varepsilon_s)L_\downarrow$). However, as the second term is usually less than ~5% of the land surface counterpart ($\varepsilon_s \sigma T_S^4$) for most land covers (Oke, 1987) it is ignored for simplicity in the development of AnOHM (cf. Appendix A).

By assuming the incoming solar radiation $K_\downarrow$ and air temperature $T_a$ follow sinusoidal forms through a day as function of the
mean value for the day (e.g. $\overline{K_\downarrow}$) (Sun et al., 2013):

$$K_\downarrow = A_K \sin(\omega t) + \overline{K_\downarrow} \tag{15}$$

$$T_a = A_T \sin(\omega t - \tau) + \overline{T_a} \tag{16}$$

and introducing the solar radiation scale:

$$A_K^* = (1 - \alpha) A_K \tag{17}$$

and longwave radiation scale (assuming $\varepsilon_a \approx \varepsilon_s \approx \varepsilon$ as a first order estimate (as AnOHM is insensitive to this parameter see section 3.2); cf. clear sky of ~0.85 (Staley and Jurica, 1972) and urban surfaces of ~0.95 (Kotthaus et al., 2014)):

$$A_T^* = \left(4\varepsilon\sigma\overline{T_a}^3 + (1 + \beta^{-1})C_h U\right) A_T = (f_L + f_T) A_T = f A_T \tag{18}$$

where $\tau$ denotes phase difference between $T_a$ and $K_\downarrow$, the $f = f_L + f_T$ consists of the longwave energy redistribution factor: $f_L = 4\varepsilon\sigma\overline{T_a}^3$ and a turbulent energy redistribution factor: $f_T = (1 + \beta^{-1})C_h U$. Linearizing the fourth-order longwave expressions of temperature at mean daily air temperature $\overline{T_a}$ (Sun et al., 2013), the values of $\overline{T_S}$ and $A_{T_S}$ are obtained:

$$\overline{T_S} = \frac{1-\alpha}{f}\overline{K_\downarrow} + \overline{T_a} \tag{19}$$

$$\begin{aligned} A_{T_S} &= \frac{fMN\sin(\tau)}{N(fM+k)\sin(\gamma) - kM\cos(\gamma)}A_T \\ &= \frac{1}{\sqrt{M_*^2 + N_*^2}}\frac{\sin(\tau)}{\sin(\gamma - \zeta)}A_T \\ &= \chi_\gamma A_T \end{aligned} \tag{20}$$

where $\zeta = \arctan(N_*/M_*)$, $\gamma = \zeta + \arctan\left(\frac{\sin(\tau)}{\cos(\tau) + A_K^*/A_T^*}\right)$, $M_* = 1 + k/(fM)$, $N_* = k/(fN)$ and $\chi_\gamma = \frac{1}{\sqrt{M_*^2 + N_*^2}}\frac{\sin(\tau)}{\sin(\gamma - \zeta)}$.

The net all-wave radiation $Q^*$ is parameterized as:

$$\begin{aligned} Q^* &= (1-\alpha)K_\downarrow + \varepsilon\sigma T_a^4 - \varepsilon\sigma T_S^4 \\ &= (1-\alpha)\left(A_K\sin(\omega t) + \overline{K_\downarrow}\right) + f_L(T_a - T_S) \\ &= c_\varphi \sin(\omega t + \varphi) + \frac{f_L}{f}(1-\alpha)\overline{K_\downarrow} \end{aligned} \tag{21}$$

where $\varphi = \arctan\left[\frac{(\chi_\gamma \sin(\gamma) - \sin(\tau))}{(fA_K^*)/(f_L A_T^*) - (\chi_\gamma \cos(\gamma) - \cos(\tau))}\right]$ and $c_\varphi = \sqrt{\left[\frac{(fA_K^*)^2}{(f_L A_T^*)^2} - (\chi_\gamma \cos(\gamma) - \cos(\tau))\right]^2 + [\beta_\gamma \sin(\gamma) - \sin(\tau)]^2}$.

**2.3 Derivation of AnOHM coefficients**

Based on the above parameterizations of $Q^*$ and $\Delta Q_S$, together with OHM for a specific surface:

$$\Delta Q_S = a_1 Q^* + a_2\frac{\partial Q^*}{\partial t} + a_3, \tag{22}$$

the coefficients can be readily derived from the parameterization in section 2.2, as:

$$a_1 = \frac{c_\eta}{c_\varphi}\cos(\eta - \varphi) \tag{23}$$

$$a_2 = \frac{c_\eta}{\omega c_\varphi}\sin(\eta - \varphi) \tag{24}$$

$$a_3 = -\frac{c_\eta}{c_\varphi} \cos(\eta - \varphi) \cdot \frac{f_T}{f} (1 - \alpha)\overline{K_\downarrow}$$

$$\qquad = -a_1 \cdot \frac{f_T}{f} (1 - \alpha)\overline{K_\downarrow} \tag{25}$$

In the densest parts of cities, the anthropogenic heat ($Q_F$) often has a large influence on the SEB that needs to be accounted for (Allen et al., 2011; Chow et al., 2014; Nie et al., 2014; Sailor, 2011). This requires the governing SEB relation (eqn 14) to be rewritten:

$$(1 - \alpha)K_\downarrow + \varepsilon\sigma T_a^4 - \varepsilon\sigma T_S^4 + Q_F = C_h U(1 + \beta^{-1})(T_S - T_a) + G_0 \tag{26}$$

Assuming $Q_F$ is diurnally invariant (as a first order estimate e.g. Best and Grimmond (2016)), the derivation (section 2.2) can be extended to include a first order estimate of $Q_F$ to obtain:

$$a_{3F} = -\frac{c_\eta}{c_\varphi} \cos(\eta - \varphi) \cdot \frac{f_T}{f} (1 - \alpha)\overline{K_\downarrow} - Q_F$$

$$\qquad = -a_1 \cdot \frac{f_T}{f} (1 - \alpha)\overline{K_\downarrow} - Q_F \tag{27}$$

where $a_{3F}$ (subscript 'F' indicates the inclusion of $Q_F$). The other two coefficients remain unchanged.

### 2.4 Physical Interpretations of AnOHM Coefficients

Based on the parameterizations of AnOHM coefficients (eqns 23, 24, 25/27), the physical interpretations can be more fully illustrated compared with OHM as follows:

   a) $a_1$ characterizes the ratio of $\Delta Q_S$ and $Q^*$ and depends on the energy scales (i.e. $c_\eta$ and $c_\varphi$) and their phase difference (i.e. $\eta - \varphi$). The energy scales, representing daily amplitudes of $\Delta Q_S$ and $Q^*$, determine the overall magnitude while the phase difference moderates the ratio value.

b) $a_2$ accounts for the temporal changes in $\Delta Q_S$ and $Q^*$ by including the principle Earth's rotation frequency $\omega$, in addition to the same determinants of $a_1$ (i.e. $c_\eta$, $c_\varphi$ and $\eta - \varphi$). The complementary sinusoidal functions, with phase difference (i.e., $\sin(\eta - \varphi)$ and $\cos(\eta - \varphi)$), in the formulations of $a_1$ and $a_2$ are inverse related with a stronger lag effect from $a_2$, and less contribution to $\Delta Q_S$ by $Q^*$ (i.e., smaller $a_1$);

   c) $a_3$ (or $a_{3F}$) indicates the baseline $\Delta Q_S$ determined by energy redistribution factors (i.e. $f_T$ and $f$) and energy inputs (i.e.

$\overline{K_\downarrow}$, and $Q_F$ if anthropogenic heat is considered) as well as $a_1$. It can be inferred from eqn 2 that the nocturnal $\Delta Q_S$ is

largely determined by $a_3$ when the absolute values and variability of $Q^*$ are small at night. A larger daytime energy input (i.e., $\overline{K_\downarrow}$, and $Q_F$ if anthropogenic heat is considered) suggests more heat released at night.

## 3 Sensitivity Analysis

Due to the complex dependence of AnOHM coefficients on surface properties and meteorological forcing (section 2.3), the
impacts of these coefficients are further assessed by a sensitivity analysis.

### 3.1 Subset Simulation

To improve the computational efficiency of undertaking Monte Carlo sensitivity analyses, subset simulation is used (Au and Beck, 2001). This is an adaptive stochastic simulation procedure with particular efficiency in analysing the short-tail of a distribution probability (while also adaptable to long-tail scenarios) (Wang et al., 2011).

If the probability that a critical response $Y$ exceeds a threshold $y$, $P(Y > y)$, a range of exceedance regions can be specified and sampled using Markov chains. Initially a direct Monte Carlo method is used to choose possible values for the parameter of interest in the anticipated range with a specified distribution (or probability distribution function, PDF) of the uncertainty. From this (level 0), the first exceedance level probability is determined, $F_1$ at which $P(Y > y_1)$. Then a Markov chain Monte
Carlo (MCMC) procedure is used to generate samples of a given conditional probability $p_0$, leading to the exceedance of $y_1$ in the earlier simulations. This procedure is repeated, for exceedance events $F_i$ at which $P(Y > y_i) = p_0^i$, $i = 1, 2, 3, ...$, until simulations reach a target exceedance probability, e.g. associated with rare events or risk analysis. Further details of this subset simulation process are provided in Wang et al. (2011).

Subset simulation efficiently generates conditional samples with Metropolis algorithms (Hastings, 1970; Metropolis et al., 1953). This is the basis of MCMC. To generate samples that successively approach a certain conditional probability, a specific Markov chain is designed with the target PDF as its limiting stationary distribution trend as its length increases. The selection of proposal distributions is the key as this controls the next sample generated based on the current one. Ideally, the distribution selection would be automatic but this has an efficiency cost relative to the robustness benefit. For the surface parameters (Table
1a) and hydrometeorological forcing (Table 1b) analyses a normal distribution PDF is used (Au and Beck, 2003; Au et al., 2007), with three conditional levels ($N_{level}$=3) and a conditional probability of $p_0 = 0.1$: i.e., at each level the highest 10% of the outputs are considered to exceed the intermediate threshold. As such, the three-level simulation can effectively capture a rare event with the target exceedance probability of $10^{-4}$ (i.e., the probability of occurrence is less than 1 in 10000) and generate appropriate samples of different conditional probabilities.


The metric $S$ (in %), used to indicate the sensitivity of the model output $Y$ to a specific uncertainty parameter $X$ (Wang et al., 2011), is:

$$S = \left| \frac{1}{N_{\text{level}}} \sum_{i=1}^{N_{\text{level}}} \frac{E[X|Y > y_i] - E[X]}{E[X]} \right| \times 100 \tag{28}$$

where $i = 1, 2, ..., N_{\text{level}}$ is the index of conditional sampling level, $E[X]$ is the expectation that the unconditional distribution of a specific uncertainty parameter $X$, while $E[X|Y > y_i]$ is the expectation of $X$ at conditional level $i$. A positive (negative) $S$ indicates an increase will lead to increase (decrease) in simulated value, hence the sign of $S$ indicates the impact of a change in parameter uncertainty. The absolute magnitude of $S$ indicates the sensitivity.

This assessment does not consider if the simulated values have low probability. Later analyses (section 4) consider the simulation results relative to observed fluxes.

**3.2 Impacts of Surface Properties**

Following the sensitivity analysis of AnOHM coefficients to the surface properties, the distributions of conditional samples for thermal conductivity $k$, bulk heat capacity $C_p$ and emissivity $\varepsilon$ are similar to the original proposal distributions (Figure 1), implying weak dependence of $a_1$, $a_2$ and $a_3$ on these properties. However, for albedo ($\alpha$) both $a_2$ and $a_3$ are sensitive but $a_1$ is not; changes in inverse Bowen ratio ($\beta^{-1}$) impact all three coefficients; and bulk transfer coefficient $C_h$ impacts $a_1$ and $a_2$, but has little effect for $a_3$.

Using $S$ (eqn 28) to quantify this, it is found that the surface properties ($k$, $C_p$ and $\varepsilon$) have less sensitivity, with less skewed conditional samples between levels, so $S$ values close to 0 (Figure 2). The $S$ of $k$ is the largest of the three. From the $S$ results for the $\alpha$ sensitivity analysis (Figure 2), it is apparent that an increase in $\alpha$ will increase $a_1$ while decreasing $a_2$ and $a_3$, whereas the reverse occurs for $\beta^{-1}$ and $C_h$ – i.e. their decreases leads to larger $a_2$ and $a_3$ values but smaller $a_1$.

From this, the links between the key surface parameters and the storage heat flux can be considered. With an increase in $\alpha$, there is reduced solar energy in the SEB - this reduces the temporal change in $\Delta Q_S$ (smaller $a_2$) and decreases the baseline value of $\Delta Q_S$ (smaller $a_3$); larger $\beta^{-1}$ indicates more available energy is dissipated by $Q_E$ than by $Q_H$, leading to decreased $T_s$ and $\Delta Q_S$ (smaller $a_1$); a smaller portion of $Q^*$ will be dissipated by $\Delta Q_S$ (smaller $a_1$) as the increased $C_h$ can facilitate the turbulent convection and thus increase the total turbulent fluxes.

**3.3 Impacts of Hydrometeorological Conditions**

Similarly, the sensitivity of AnOHM to hydrometeorological variables is explored (Figure 3). The air temperature (range, mean) and water flux density related variables (i.e. $A_T$, $\overline{T_a}$ and $W$) have minimal influence on the skewness of the conditional samples.

In contrast, the incoming shortwave (solar) radiation (range, mean) and wind related variables (i.e. $A_K$, $\overline{K_\downarrow}$ and $U$) and the phase lag $\tau$ between $K_\downarrow$ and $T_a$ have large impacts. In terms of the greatest impact on the coefficients ($a_1$, $a_2$ and $a_3$): $A_K$ and $U$ influences $a_1$, $\tau$ impacts $a_2$, and $a_3$ responds more to $A_K$ and $\overline{K_\downarrow}$ than the other variables.

Variables that strongly modulate the interactions between $\Delta Q_S$ and $Q^*$ can be informed by the $S$ results (Figure 4). For instance, a greater range in $K_\downarrow$ (i.e. larger $A_K$) will occur with larger energy input from solar radiation, leading to stronger heating of the near-surface atmosphere and a smaller portion to $\Delta Q_S$ (smaller $a_1$) but higher baseline $\Delta Q_S$ (larger $a_3$). This is consistent with a reduction in $\overline{K_\downarrow}$ having a decrease in $a_3$. The temporal change in $\Delta Q_S$ is highly correlated with the change in $\tau$, an increase in which implies a slower response of the surface to solar radiation and an overall decrease in $\Delta Q_S$ (smaller $a_1$, $a_2$ and $a_3$). The

greater sensitivity to $\tau$ of $a_2$ is a key part of the original hysteresis nature of the heating/cooling of a surface. The sensitivity responses of $a_1$, $a_2$ and $a_3$ to $U$ are very consistent with those to $C_h$, suggesting the similar pathway that turbulent fluxes (i.e. $Q_H$ and $Q_E$) modulate $\Delta Q_S$. As $W$ mostly influences the heat conduction-diffusion in the underlying surface as thermal properties (i.e. $C_p$ and $k$), less dependence is observed on it. This is similar with $C_p$ and $k$.

## 4 Model Evaluation

In this section, the actual ability of AnOHM to determine the storage heat flux relative to observations is evaluated using 30 min observations from five sites of different land use/covers (Table 3). The measurements include turbulent sensible and latent fluxes, along with incoming and outgoing shortwave and longwave radiation and basic meteorological variables (refer to Kotthaus and Grimmond (2014b; 2014a), Klazura et al. (2006), Coulter et al. (2006), Goulden et al. (2006), Scott et al. (2009), and Luo et al. (2007) for details). Anthropogenic heat flux $Q_F$ at the urban site (i.e., UK-Ldn) is estimated using the GreaterQF

model (Iamarino et al., 2011); the heat storage flux $\Delta Q_S$ is thus estimated as the modified residual of urban energy balance as $\Delta Q_S = Q^* + 0.75 Q_F - 1.2(Q_H + Q_E)$ (Kotthaus and Grimmond, 2014a; 2014b), which is then used in this evaluation. A similar approach for estimating $\Delta Q_S$ (i.e., residual of surface energy balance, $\Delta Q_S = Q^* + Q_F - (Q_H + Q_E)$) is applied at other non-urban sites but with $Q_F = 0$.

AnOHM is first calibrated with observations under sunny conditions, when the assumptions of AnOHM are best satisfied (i.e., diurnal cycles of $K_\downarrow$ and $T_a$ follow sinusoidal forms), to obtain surface properties required by AnOHM (Table 4). As the Bowen ratio $\beta$ varies daily and monthly (Kotthaus and Grimmond, 2014a; 2014b), $\beta$ is either determined as the daily value if available, or based on the observation-based monthly climatology (Table 4). The seasonality in albedo $\alpha$ is accounted for also by using its monthly climatology (Table 4). AnOHM is driven by atmospheric forcing (i.e., $K_\downarrow$, $T_a$, and $U$) and/or their derived scales

($A_K$, $\overline{K_\downarrow}$, $A_T$, $\overline{T_a}$ and $\tau$) to generate the OHM coefficients (i.e., $a_1$, $a_2$ and $a_3$, cf. Figure 5), from which the net heat storage flux $\Delta Q_S$ can be predicted (Figure 6) using the observed $Q^*$ with equation 2.

To examine the seasonality of the OHM coefficients, rather than the daily variations in hydrometeorological forcing, LOESS (LOcally wEighted Scatter-plot Smoother, Cleveland and Devlin (1988)) curves are obtained to filter out day-to-day variations in the OHM coefficients (cf. Appendix B for a direct comparison of these coefficients by different modelling and observational regression approaches). Intra-annual variations are found in all the three OHM coefficients (Figure 5), indicating the strong impact of seasonality of meteorological conditions. These controls, as indicated by eqns 23–25(27), are complex and will vary with local conditions. For instance, comparison of OHM coefficients between the AnOHM predictions (LOESS fitted solid lines in Figure 5) and observations at an asphalt road site in Alland, Austria reported in Anandakumar (1999) (empty squares in Figure 5) demonstrates differences in $a_1$ (Figure 5a) and $a_2$ (Figure 5b) but general similarity in $a_3$ (Figure 5c). Compared to $a_1$ and $a_2$, it is noteworthy that, in addition to the $S$ results (cf. Figure 4) given the more explicit mechanism by which the atmospheric conditions moderate $a_3$ (cf. equation 25/27), such seasonality in $a_3$ is predicted by AnOHM, and evident in the observations (Figure 5c, also Ward et al. (2013)). Larger $\overline{K_\downarrow}$ in warm seasons (May–September) will lead to smaller $a_3$ (cf. equation 25/27) and *vice versa*.

The AnOHM simulated and observed $\Delta Q_S$ agree well at the five different land cover sites, with RMSE values of ~30 W m$^{-2}$. For comparison purposes, it is noted that the urban land surface model comparison (Best and Grimmond, 2015; Grimmond et al., 2011), found $\Delta Q_S$ to be the most poorly represented among all the SEB components with the best RMSE values of 53 W m$^{-2}$ (Lipson et al., 2016). Although the much smaller $\Delta Q_S$ RMSE obtained by AnOHM uses a prescribed Bowen ratio in the offline evaluation, such improvement indicates the ability of AnOHM to simulate a more consistent $\Delta Q_S$ with observations. Compared with OHM predictions (orange lines in Figure 6), AnOHM (blue lines in Figure 6) better reproduces the seasonality in $\Delta Q_S$ but gives larger bias at two sites with natural land covers (i.e., US-SRM and US-SO4). This can be attributed to the overestimates of nocturnal $\Delta Q_S$ by AnOHM. Overall, the evaluation demonstrates good performance of AnOHM in predicting the long-term $\Delta Q_S$ with clear seasonality reproduced across a wide range of surface types.

**5 Discussion and Concluding Remarks**

In this study, an Analytical Objective Hysteresis Model (AnOHM) is developed to obtain OHM coefficients across a wide range of surface and meteorological conditions and to improve physical understanding of the interactions between $\Delta Q_S$ and $Q^*$. The sensitivity of AnOHM to surface properties and hydrometeorological conditions are analysed through Monte Carlo based Subset Simulations (Au and Beck, 2001). The results highlight the importance of the albedo, the Bowen ratio and the bulk transfer coefficient, and the importance of solar radiation and wind speed in regulating the heat storage. The importance of albedo in modulating the heat storage was also found by Wang et al. (2011) as well using the same Subset Simulation approach with the urban canopy model (UCM, details refer to Kusaka et al. (2001)). This demonstrates the consistency in heat

storage modelling between AnOHM and UCM. From the sensitivity results, variations in OHM coefficients of a similar size may arise from either surface property parameters or hydrometeorological forcing that are associated with the same physical processes (cf. bulk transfer coefficient $C_h$ in Figure 2 and wind speed $U$ in Figure 4). This suggests the ability of AnOHM in representing physical processes. An offline evaluation of AnOHM using flux observations from five sites with different land covers demonstrates its ability to predict the intra-annual dynamics of OHM coefficients and shows good agreement between simulated and observed storage heat fluxes. In particular, the seasonality in the OHM coefficient $a_3$ observed in a previous study (Anandakumar, 1999) is well predicted by AnOHM.

The limitations of AnOHM are important to consider. First, given the assumption that the incoming solar radiation $K_\downarrow$ and air temperature $T_a$ diurnal cycles are sinusoidal, optimal performance of AnOHM occurs under clear sky conditions. The current parameterisations of $K_\downarrow$ and $T_a$ within AnOHM only consider the harmonics of principal frequencies for formulation simplicity. More frequencies may potentially resolve more realistic diurnal variations in $K_\downarrow$ and $T_a$. As the reflected part of $L_\downarrow$ (i.e., $(1 - \varepsilon_s)L_\downarrow$ ) is assumed negligible and similar emissivity values are assumed for sky and land surface (i.e., $\varepsilon_s \approx \varepsilon_a \approx \varepsilon$), the outgoing longwave radiation is underestimated. These simplifications greatly facilitate the AnOHM formulation without qualitatively changing the final results as the sensitivity analyses (cf. minimal $S$ values for $\varepsilon$ in Figure 2) demonstrate. The inclusion of water flux density $W$ equips AnOHM with an ability to investigate the hydrological impacts of the underlying surface on land-atmosphere interactions. However, estimation of $W$ remains challenging (Wang, 2014) and the resulting uncertainty in the final results warrants caution in conducting simulations over land covers with strong soil moisture dynamics (e.g., grassland with high soil moisture under clear sky condition).

Despite these limitations, AnOHM does permit improved modelling of the surface energy balance through its physically-based parameterization scheme for storage heat flux $\Delta Q_S$. Compared to OHM, AnOHM has the benefit of allowing $\Delta Q_S$ to be simulated for land covers for which coefficients are not available and to allow for seasonal variability to be accounted for. As AnOHM shares similar hydrometeorological forcing inputs (i.e., $K_\downarrow$, $T_a$ and $U$) to other land surface models (LSMs), it can potentially be used within in LSMs to estimate $\Delta Q_S$, or if turbulent fluxes are included be a complete LSM. The overall improvements from adopting AnOHM in modelling land surface processes will be presented in forthcoming work in the SUEWS-AnOHM framework.

**Code availability**

The Fortran source code for AnOHM can be obtained from the corresponding authors upon request.

**Appendix A: Rationale for A Simplified Formulation of Outgoing Longwave Radiation**

In the formulation of outgoing longwave radiation $L_\uparrow$, a simplified form (i.e., $\varepsilon_s \sigma T_s^4$) is used for AnOHM as eqn 10 by ignoring the reflected part of $L_\downarrow$ (i.e., $(1 - \varepsilon_s)L_\downarrow$). The rationale for such simplification is explained that given $\varepsilon_s$ is usually larger than 0.9, $(1 - \varepsilon_s)L_\downarrow$ contributes a relatively small portion to the total longwave component (Oke, 1987) and omission of this part

is well accepted in the parameterization of upwelling longwave radiation for land surface modeling across various land covers (Bateni and Entekhabi, 2012; Lee et al., 2011).

Using the parameterisation of incoming longwave radiation in the AnOHM framework (i.e., $L_\downarrow = \varepsilon_a \sigma T_a^4 \approx \varepsilon_s \sigma T_a^4$), we conduct a sensitivity analysis of the ratio between the ignored part (i.e., $(1 - \varepsilon_s)L_\downarrow$) and total upwelling longwave radiation (i.e., $\varepsilon_s \sigma T_s^4 + (1 - \varepsilon_s)L_\downarrow$) at a constant air temperature of 20 °C and find this ratio is generally less than 5% given $\varepsilon_s$ ranges

between 0.90 and 0.99 (Figure A1).

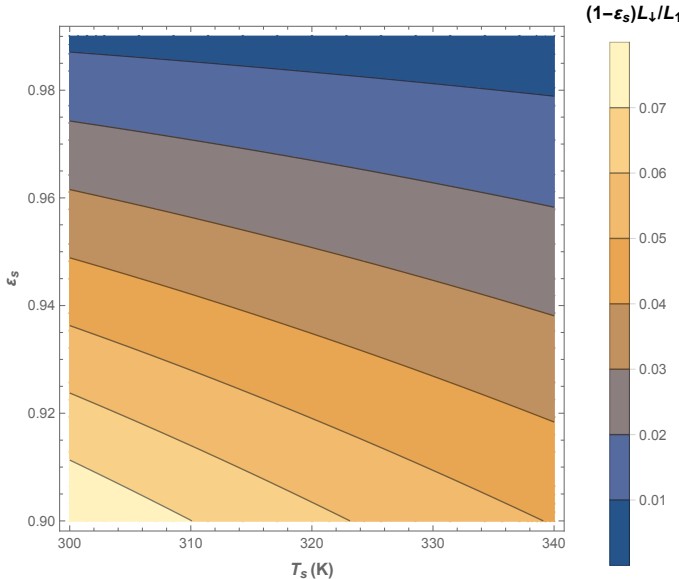

**Figure A1** Ratio between reflected (i.e., $(1 - \varepsilon_s)L_\downarrow$) and total upwelling longwave radiation (i.e., $\varepsilon_s \sigma T_s^4 + (1 - \varepsilon_s)L_\downarrow$) at a constant air temperature of 20 °C.

Moreover, if $(1 - \varepsilon_s)L_\downarrow$ is included in the net longwave radiation, the induced effect can be incorporated into a modified sky emissivity $\varepsilon_a' = \varepsilon_s \varepsilon_a$ as follows:

$$
\begin{aligned}
L_{net} &= L_\downarrow - L_\uparrow \\
&= L_\downarrow - (\varepsilon_s \sigma T_s^4 + (1 - \varepsilon_s)L_\downarrow) \\
&= \varepsilon_s L_\downarrow - \varepsilon_s \sigma T_s^4 \\
&= \varepsilon_s \varepsilon_a \sigma T_a^4 - \varepsilon_s \sigma T_s^4 \\
&= \varepsilon_a' \sigma T_a^4 - \varepsilon_s \sigma T_s^4
\end{aligned}
$$

Then by assuming $\varepsilon \approx \varepsilon_a' \approx \varepsilon_s$, the derivation following equation 18 still holds. The sensitivity analysis suggests that the derived coefficients are insensitive to $\varepsilon$ (cf. $S$ for $\varepsilon$ in Figure 2).

As such, we deem the omission of $(1 - \varepsilon_s)L_\downarrow$ will not qualitatively change the results of this work.

**Appendix B: Comparison in OHM Coefficients between Different Modelling Approaches and Observation Regression**

5     The comparison in OHM coefficients by different modelling and observational regression approaches (Figure A2) indicate AnOHM generally follows the results by observation regression, whereas the typical coefficient values adopted by OHM do not.

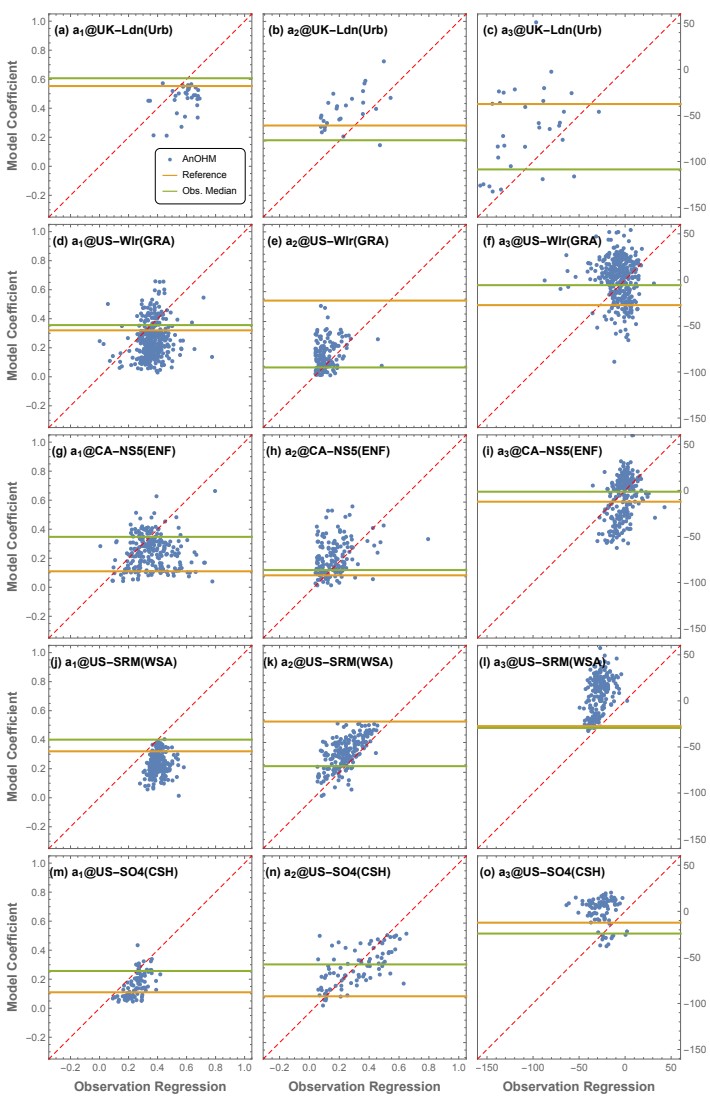

**Figure A2** Comparison of OHM coefficients (left, central and right columns for $a_1$, $a_2$ and $a_3$, respectively) between different modelling approaches and observation regression at five sites: UK-Ldn (a, b, c), US-Wlr (d, e, f), CA-NS5 (g, h, i), US-SRM (j, k, l) and US-SO4 (m, n, o). The blue dots denote the paired values between AnOHM and observation regression. The orange lines represent the reference value used in OHM simulations for land covers of grass and tree (Grimmond and Oke, 1999), whereas the green lines shows median values derived from results by observation regression at corresponding sites.

**Acknowledgement**

Funding is acknowledged from Met Office/Newton Fund CSSP- China (SG), National Science Foundation of China (51679119, TS), and U.S. National Science Foundation (CBET-1435881, ZHW). The authors thank Professor Ivan Au (University of Liverpool) for providing the Subset Simulation package. The authors acknowledge the large number of people who have contributed to the data collection, the agencies that have provided sites and the agencies that have funded the research at the individual sites. The US Department of Energy's Office of Science funded AmeriFlux data (ameriflux-data.lbl.gov) are from: US-Wlr (PIs: Dr David Cook and Dr Richard L. Coulter), CA-NS5 (PI: Dr Mike Goulden), US-SRM (PI: Dr Russell Scott) and US-SO4 (PI: Dr Walt Oechel, funded by San Diego State University and SDSU Field Stations Program). The London data are supported by NERC ClearfLo (NE/H003231/1), NERC/Belmont TRUC (NE/L008971/1), EUf7 BRIDGE (211345), H2020 UrbanFluxes (637519), King's College London and University of Reading. In particular, the authors thank Dr Simone Kotthaus (University of Reading) for her detailed preparation of the UK-Ldn site data. For access to the UK-Ldn site data, please contact c.s.grimmond@reading.ac.uk.

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

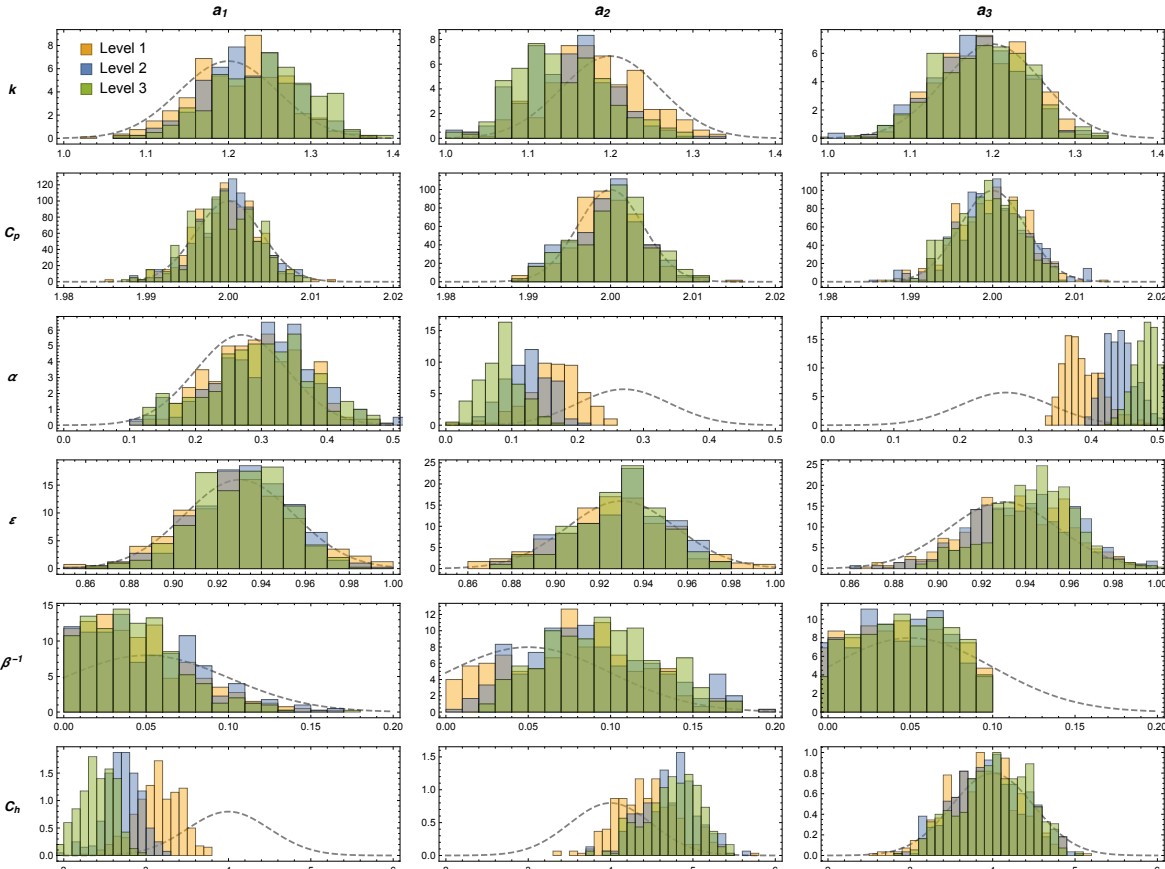

**Figure 1 Histograms of conditional samples at different conditional levels for surface property parameters (rows from top: thermal conductivity $k$ in W m$^{-1}$ K$^{-1}$, heat capacity $C_p$ in MJ m$^{-3}$ K$^{-1}$, albedo $\alpha$, emissivity $\varepsilon$, inverse Bowen ratio $\beta^{-1}$ and bulk transfer coefficient $C_h$ in J m$^{-3}$ K$^{-1}$) with AnOHM coefficients as the model output (columns from left: $a_1$, $a_2$ and $a_3$). Each subplot x-axis is the parameter value and y-axis is the PDF value. The original proposal distribution (dashed line) and simulation levels (different colours) are shown.**

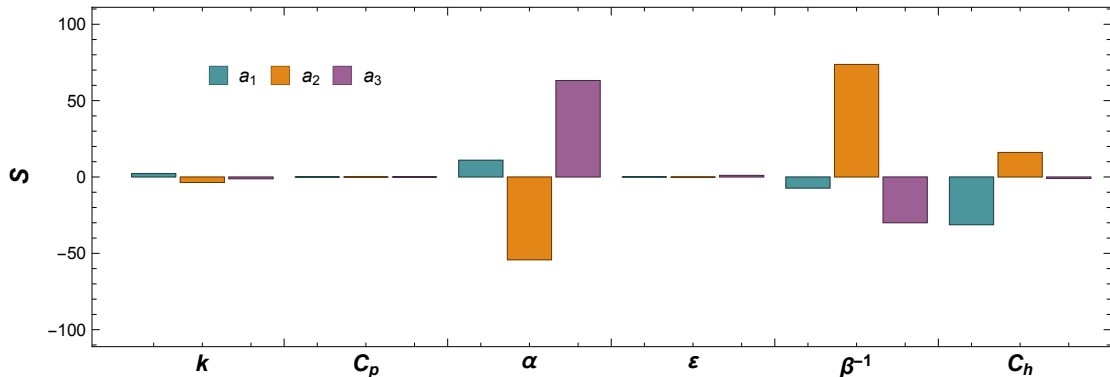

**Figure 2 Relative variation in sensitivity (*S*, %, equation 28) to surface parameters. See Figure 1 for further details.**

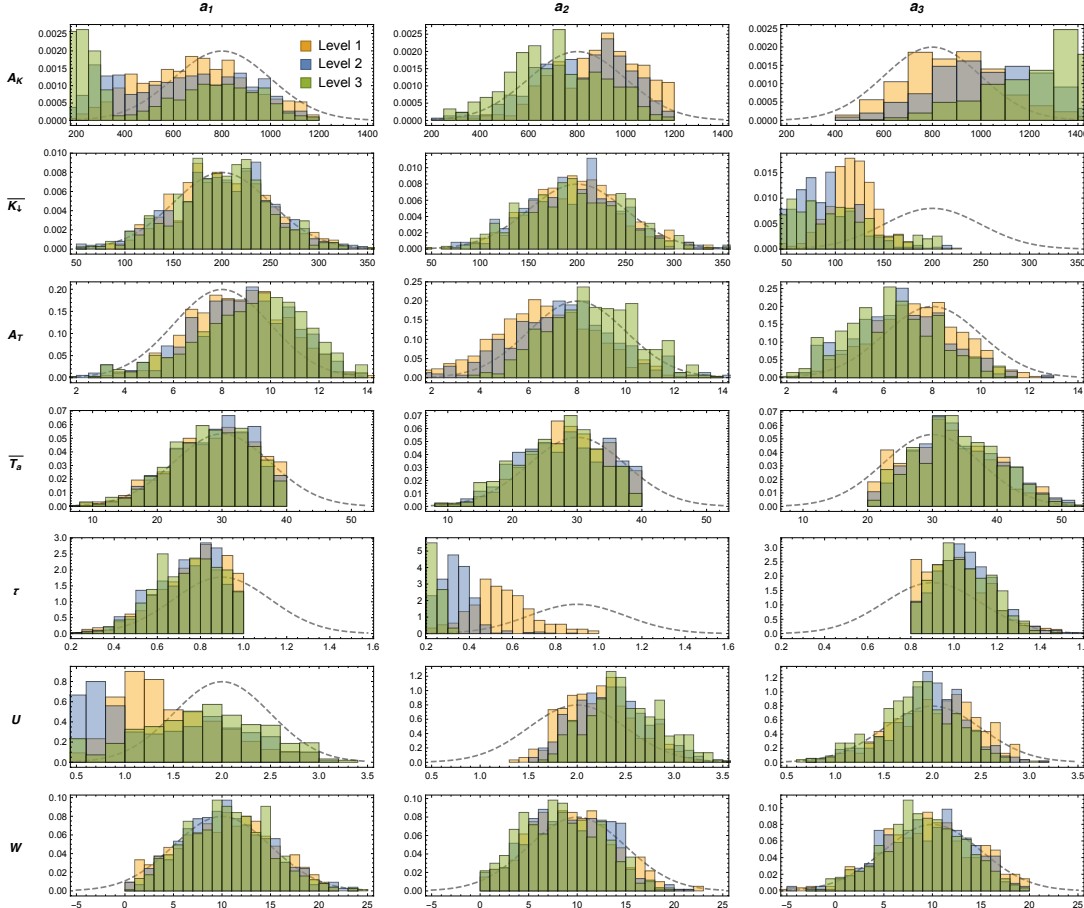

**Figure 3** Histograms of conditional samples at different conditional levels for ambient forcing parameters (rows from top: incoming solar radiation amplitude $A_K$ in W m$^{-2}$ and its daytime mean $\overline{K_\downarrow}$ in W m$^{-2}$, air temperature amplitude $A_T$ in °C and its daily mean $\overline{T_a}$ in °C, the phase lag $\tau$ in rad between $K_\downarrow$ and $T_a$, wind speed $U$ in m s$^{-1}$ and water flux density $W$ in m s$^{-1}$) with AnOHM coefficients as the model output (columns from left: $a_1$, $a_2$ and $a_3$). As Fig. 1.

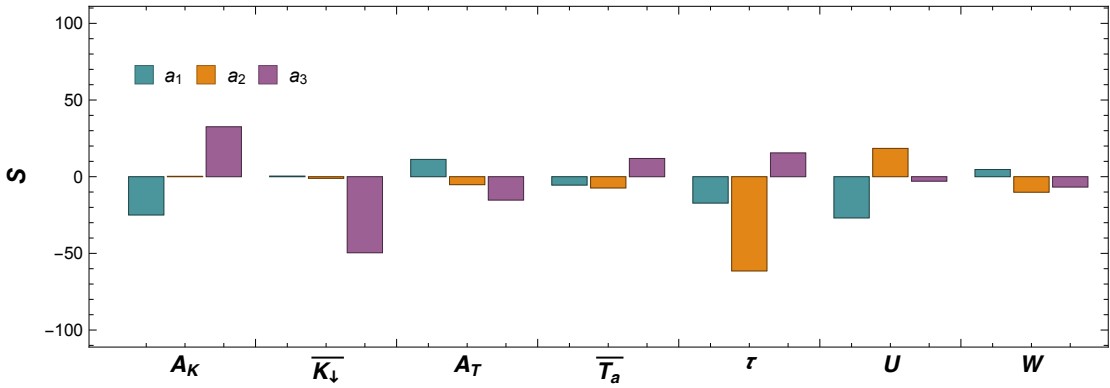

**Figure 4** Relative variation in sensitivity (*S*, %, equation 28) to forcing parameters. See Figure 3 for further details.

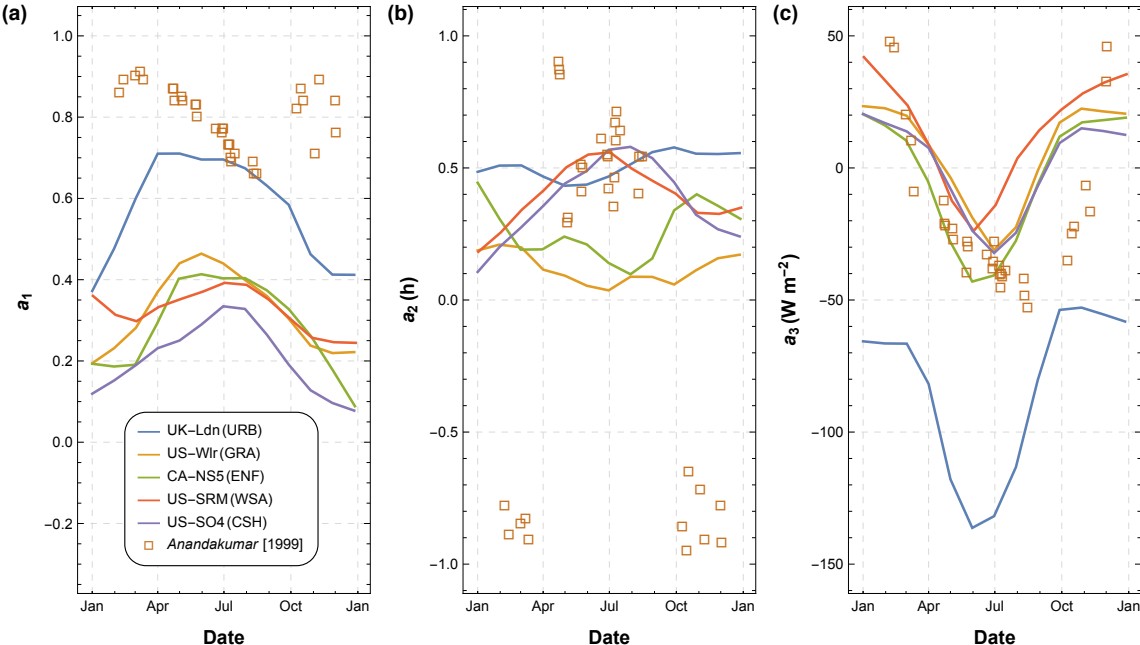

**Figure 5** Intra-annual variations of OHM coefficients: (a) $a_1$, (b) $a_2$ and (c) $a_3$. LOESS fits (solid lines) through the daily values predicted by AnOHM and daily values (squares) measured at an asphalt road site (Anandakumar 1999) are shown. The LOESS (Cleveland and Devlin 2012) fitting is a locally weighted polynomial regression approach.

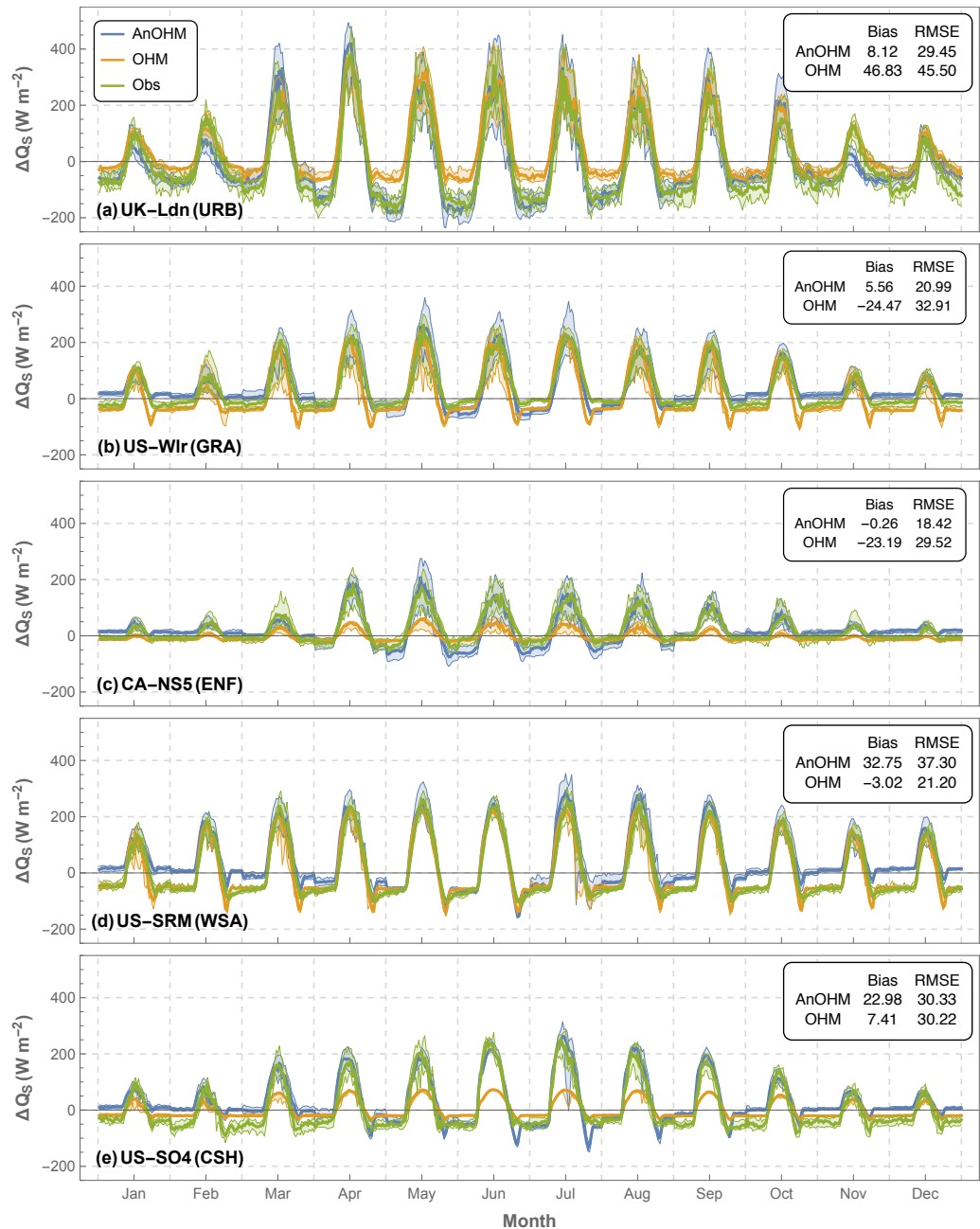

**Figure 6 Monthly median (line) diurnal cycles and interquartile (shaded) values of $\Delta Q_S$ for AnOHM predictions (blue), OHM predictions (orange) and observations (green) at (a) UK-Ldn (URB), (b) US-Wlr (GRA), (c) CA-NS5 (ENF), (d) US-SRM (WSA), and (e) US-SO4 (CSH) (see Table 2 for site information). Statistics include average bias and RMSE (W m$^{-2}$). The OHM coefficients $a_1$, $a_2$ and $a_3$ used for different land covers are: 0.553, 0.303 and -37.6 at the urban site (UK-Ldn) (Ward et al., 2016), 0.32, 0.54 and -27.4 at the grass covered sites (US-Wlr and US-SRM) (Grimmond and Oke, 1999), and 0.11, 0.11 and -12.3 at the forest covered sites (CA-NS5 and US-SO4) (Grimmond and Oke, 1999).**

**Table 1 Range of values used as basis for the sensitivity analysis (a) surface parameters and (b) meteorological variables. All are assumed to have normal PDF. Values of surface parameters are based on values reported in Stull (Stull, 1988).**

| Parameter/Variable | | Unit | Min | Max | Mean | Standard deviation |
|---|---|---|---|---|---|---|
| *(a) Surface* | | | | | | |
| thermal conductivity | $k$ | W m$^{-1}$ K$^{-1}$ | 0 | 3 | 1.2 | 0.1 |
| bulk material heat capacity | $C_p$ | MJ m$^{-3}$ K$^{-1}$ | 0 | 4 | 2.0 | 0.04 |
| albedo | $\alpha$ | -- | 0 | 1 | 0.27 | 0.07 |
| emissivity | $\varepsilon$ | -- | 0.8 | 1.0 | 0.93 | 0.025 |
| midday* mean Bowen ratio (inverse) | $\beta^{-1}$ | -- | 0 | 20 | 0.05 | 0.05 |
| bulk transfer coefficient | $C_h$ | J m$^{-3}$ K$^{-1}$ | 0 | 8 | 4 | 0.5 |
| *(b) Hydrometeorological* | | | | | | |
| Amplitude or range of the daily incoming shortwave radiation | $A_K$ | W m$^{-2}$ | 0 | 1200 | 800 | 200 |
| Mean daytime incoming shortwave radiation | $\overline{K_\downarrow}$ | W m$^{-2}$ | 0 | 500 | 200 | 50 |
| Amplitude or range of the daily air temperature | $A_T$ | ºC | 0 | 15 | 8 | 2 |
| Mean daily air temperature | $\overline{T_a}$ | ºC | 0 | 40 | 30 | 7.5 |
| Phase lag between radiation and air temperature | $\tau$ | rad | 0 | $\pi/2$ | $\pi/4$ | $\pi/10$ |
| Mean daytime wind speed | $U$ | m s$^{-1}$ | 0 | 4 | 2 | 0.5 |
| Mean daily water flux density | $W$ | $10^{-7}$ m$^3$ s$^{-1}$ m$^{-2}$ | 0 | 100 | 10 | 5 |

* midday period: 1000–1400 local standard time.

5   **Table 2 Characteristics of the flux towers at the study sites.**

| Site | UK-Ldn | US-Wlr | CA-NS5 | US-SRM | US-SO4 |
|---|---|---|---|---|---|
| Location | 51.50º N 0.12º W | 37.52º N 96.86º W | 55.86º N 98.49º W | 31.82º N 110.87º W | 33.38º N 116.64º W |
| Land cover classification | Urban/ Built-up | Grassland | Evergreen Needleleaf Forest | Woody Savannas | Closed Shrublands |
| Land cover code | URB | GRA | ENF | WSA | CSH |
| Study Year | 2011 | 2003 | 2004 | 2004 | 2005 |
| Reference | Kotthaus and Grimmond (2014a; 2014b) | Klazura et al. (2006), Coulter et al. (2006) | Goulden et al. (2006) | Scott et al. (2009) | Luo et al. (2007) |

**Table 3 Surface properties used in AnOHM simulation for the study sites based on calibration. The values of $\alpha$ and $\beta$ are monthly climatology from January to December and are used when observations are not available (see Table 1 for notation definition).**

| Parameter | Unit | Site | | | | |
|---|---|---|---|---|---|---|
| | | UK-Ldn | US-Wlr | CA-NS5 | US-SRM | US-SO4 |
| $k$ | W m$^{-1}$ K$^{-1}$ | 2.8 | 0.43 | 0.51 | 0.41 | 0.56 |
| $C_p$ | MJ m$^{-3}$ K$^{-1}$ | 2.4 | 0.31 | 0.36 | 0.56 | 0.27 |
| $\alpha$ | -- | 0.24,0.24, 0.22,0.20, 0.14,0.13, 0.12,0.14, 0.18,0.24, 0.24,0.18 | 0.29,0.29, 0.17,0.18, 0.18,0.12, 0.11,0.10, 0.19,0.13, 0.24,0.35 | 0.30,0.29, 0.22,0.15, 0.10,0.10, 0.10,0.11, 0.22,0.24, 0.28,0.30 | 0.13,0.17, 0.16,0.14, 0.13,0.12, 0.13,0.15, 0.14,0.19, 0.13,0.18 | 0.22,0.11, 0.11,0.10, 0.11,0.10, 0.10,0.10, 0.11,0.10, 0.17,0.24 |
| $\varepsilon$ | -- | 0.92 | 0.93 | 0.95 | 0.95 | 0.92 |
| $\beta$ | -- | 6.1, 5.1, 8.3, 7.9, 5.4, 3.9, 5.3, 4.2, 5.2, 4.3, 4.8, 3.2 | 2.9, 0.8, 7.6, 2.7, 0.3, 0.3, 0.3, 0.8, 0.5, 0.7, 2.3, 2.3 | 6.1, 6.0, 8.7, 8.0, 1.9, 1.6, 0.7, 0.7, 1.3, 1.4, 3.1, 8.0 | 1.9, 5.5, 3.3, 2.0, 10.1, 9.7, 2.0, 0.9, 3.0, 4.3, 10.0, 3.3 | 1.5, 1.4, 1.9, 3.0, 1.4, 1.4, 2.1, 1.2, 2.8, 1.9, 2.1, 4.1 |
| $C_h$ | J m$^{-3}$ K$^{-1}$ | 4.3 | 1.9 | 5.1 | 3.6 | 3.9 |

