# Peer review of "The Analytical Objective Hysteresis Model (AnOHM v1.0)"

_Geoscientific Model Development, 2016_

## Referee Comment (RC1) · Anonymous Referee #1 · 25 Jan 2017

The article entitled as "The Analytical Objective Hysteresis Model (AnOHM v1.0): Methodology to Determine Bulk Storage Heat Flux Coefficients" currently in submission to the journal of Geoscientific Model Development, developed an Analytical Objective Hysteresis Model (AnOHM) and performed sensitivity analysis of AnOHM to surface properties and hydrometeorological conditions. The results show that the offline evaluation of AnOHM for five different land convers generated good performances. The AnOHM improved modelling land surface processes. I recommend acceptance after revision. I am supplying comments, both general and specific as follows:

General comments:

1. This paper is potential and well organized, but the language proficiency still needs to be polished by Professor Sue Grimmond who is the fourth author and a native English speaker.

2. It will be great if authors compare their own results with the previous work.

3. This paper is lacking a formal Discussion section. I suggest the authors develop this section.

Abstract:

1. Line 14, suggest change "hampers application" to "hampers its application".

2. Line 15, change "1-dimensional" to "one-dimensional".

3. Lines 18-19, "From this albedo, Bowen ration and bulk transfer coefficient, solar radiation and wind speed are identified as being critical." I strongly recommend the authors revise this statement.

4. Line 21, change "OHM coefficients to" to "OHM coefficients".

1 Introduction

1. Page 2, Lines 9-10. "The volume of interest extends from the top of the roughness sub-layer to the depth in the ground where the vertical net heat conduction is zero on a daily basis (see Figure 2 in Masson et al., 2002)." The statement is contrary to its former statement.

2. Page 2, Lines 11-16. It is recommended to add the references at where is appropriate, such as after "e.g. 5%", and after "the term becomes much more significant",

3. Page 2, Lines 19-33, Page 3, Lines 1-17. What are the disadvantages and advantages of OHM compared with the other techniques to determine the storage heat flux? The parts of listed different techniques are verbose (Page 2, Lines 22-33).

4. Page 3, Line 3. How to determine a1, a2 and a3 by observations?

5 Page 3, Lines 14-16. Suggest change the statement "Although, Gao et al. (2003; 2008) solved the 1-dimensional advection-diffusion equation of coupled heat and liquid water transport to explore the physical relation of OHM coefficients *a1* and *a2* to the phase lag between ΔQS and Q*," to "Although, the one-dimensional advection-diffusion equation of coupled heat and liquid water transport equation was solved by Gao et al. (2003, 2008), and the solution was used to explore the physical relation of OHM coefficients a1 and a2 to the phase lag between $\Delta Q_S$ and Q* (Gao et al., 2010), ".

6. Page 3, Lines 19-26. What will be done and some results are mixed together. It is recommended that the authors revise those statements.

2 Model Development

1. Page 4. *t* should be defined below equation (3).

2. Page 4. It is strongly recommended putting equation (7) before equation (4), as "The steady-periodic solution of equation (3) with boundary condition, Ts=A_Ts sin(wt-Y)+Ts_aver  (4)".

3. Page 5. Albedo should be defined below equation (14).

4. Page 5, Line 9. It is recommended adding references or state the reason that it is reasonable to assume the incoming solar radiation and air temperature follow sinusoidal forms through a day as function of the mean value for the day.

5. Page 5, equation (8). Where is the term of longwave radiation from soil surface in the longwave radiation scale (equation (8))?

6. Page 7, Section 2.4. I strongly recommend adding statements about the advance of the AnOHM coefficients compared with the previous OHM coefficients. Based on the abstract, to enhance physical interpretations of the OHM coefficients is one of the paper's goals.

3 Sensitivity Analysis

1. Page 8, Line 20. "Stull, 1998" should be placed at "(Table 1a, based on values reported in Stull (1982))".

2. Page 9, Lines 15-17.  Based on the statement "A positive (negative) *S* indicates an increase will lead to increase (decrease) in simulated value",  an increase in albedo will increase $a_1$ and

$a_3$ while decrease $a_2$. Because Figure 2 shows that the S of $a_1$ and $a_3$ are positive and the S of $a_2$ is negative. It is strongly recommended double-checking the other statements for surface properties and the statements for hydrometerrolgical forcing parameters.

3. It may be interesting to compare the S of surface properties and hydrometerrolgical forcing parameters.

4. It is recommended comparing the results of sensitivity analysis to previous works.

4 Model Evaluation

1. What does the ability of AnOHM to capture intra-annual dynamics $\Delta Q_S$ impact its simulation $\Delta Q_S$?

2. It is recommended comparing the results of sensitivity analysis to previous works.

5 Concluding Remarks

1. Authors should be sure to inform the reader of what may be lacking in the study as well as needs for future work.

2. It is recommended to add statement that how the current work actually advances science.

---

## Referee Comment (RC2) · Anonymous Referee #2 · 11 Feb 2017

Review of the manuscript gmd-2016-300 The Analytical Objective Hysteresis Model (AnOHM v1.0): Methodology to Determine Bulk Storage Heat Flux Coefficients

Author(s): Ting Sun et al.

Summary This paper extends the well-known OHM model by including an analytical solution of the advection diffusion equation which is subsequently used to study the uncertainty and parameter sensitivities a1, a2, a3 in the OHM model using a Monte-Carlo analysis. In principle the study results are very welcome in the literature since the parameter estimation of a1, a2, and a3 are challenging and further detail is needed for successful application in a myriad of cities. However, I have a number of concerns with the paper that makes me recommend major revisions for this paper

[Figure]

Recommendation: major revisions needed

Major comments: 1. My main concern with the paper is the readability of the paper. In general the paper lacks a justification of the utilized methodologies (especially the parameter estimation, LOESS method etc) and complete description of these method. In terms of style, the paper reads a bit as a flood on information on equations and parameters, but a real interpretation of the results is missing. Overall as a reader I get too much a feeling that the whole paper provides a black box approach.

2. Interpretation: The followed approach provides new values and uncertainties in the parameter values of the OHM model. However, the paper does not reach a level beyond these parameter values. I think the reader expects more interpretation on the various parameter values and how much it would change the surface energy balance as a whole by the new information at hand. Moreover, the bias and RMSE are still quite high for some of the presented sites. I miss an outlook on how the authors will further address this, or any hypothesis behind these biases.

3. The paper is missing a discussion section. The authors can be more critical towards their results, the influence of certain assumptions made in the analysis on the results (e.g. assuming e=ea=es =0.85). Moreover AnOHM should outperform the original OHM, but this is not shown.

4. In equations (10) and (26) the upwelling component $e_s*L_{down}$ is missing. How does this missing component affect the paper's results and parameter sensitivities, especially to es?

5. Equation 21, first line: I have the impression the 4's should be removed (or the last two terms should be replaced by $4*sigma*eT^3(Ts-Ta)$).

6. P11, ln 15: I find the hit rate not a good metric to evaluate this model, at least not if presented as the only metric. In terms of contingency tables, the hit rate should always be presented together with the false-alarm rate, and preferably with an critical success

index or a threat score.
* * *

---

## Referee Comment (RC3) · Anonymous Referee #3 · 14 Feb 2017

General comments:

Urban heat storage ($\Delta Q_S$) is a large component of the urban surface energy balance, and it can be up to 30-40% of net radiation. This paper presents a parameterized approach of Objective Hysteresis Model (OHM) coefficients, based on 1-D advection-conduction equation. That's an improvement comparing to original OHM. In addition, it also gives sensitivity analysis and model evaluation. It will be very important for urban surface energy balance study. I recommend acceptance after minor revision.

Specific comments:

(1) Page 1, line 14, it is recommended describing OHM limitation more clearly.

(2) Please check Eq.27 in page 7. Dose it maybe

$$a_{3F} = -a_1 \frac{f_T}{f}(1-\alpha)\overline{K_\downarrow} - a_1 Q_F \quad ?$$

Based on Eq.22, $\Delta Q_S = a_1(Q^* + Q_F) + a_2 \dfrac{\partial(Q^* + Q_F)}{\partial t} + a_{3F}$ when $Q_F$ is

included. With the assumption that $Q_F$ is diurnal invariant,

$$\Delta Q_S = a_1(Q^* + Q_F) + a_2 \frac{\partial Q^*}{\partial t} + a_{3F}$$
$$= a_1 Q^* + a_2 \frac{\partial Q^*}{\partial t} + a_1 Q_F + a_{3F}$$

So $a_3 = a_1 Q_F + a_{3F}$, and $a_{3F} = a_3 - a_1 Q_F = -a_1 \dfrac{f_T}{f}(1-\alpha)\overline{K_\downarrow} - a_1 Q_F$.

(3) In page 10, a greater in incoming solar radiation ($K\downarrow$) will lead to smaller $\Delta Q_S$, why? In general, net radiation mostly depends on $K\downarrow$, and the larger $K\downarrow$, the larger net radiation which will lead to larger $\Delta Q_S$.

(4) In Figure 5, the blue solid line (URB) is large differently from other lines in (a) and (c). Based on Figure 5a, the $\Delta Q_S$ can be up to 70% of net radiation, it's too large to believe. In addition, there's also large difference between simulation and observation in Figure 5a, 5b. Please explain them.

---

## Editor Comment (EC1) · C. van Heerwaarden (Editor) · 2 Mar 2017

With three reviews posted, I suggest the author's reply to the reviewers' comments. I agree with the judgment of the second reviewer that this paper requires major revisions. The paper has a complete description, but it could improve in clarity as well as in organization. Following the detailed comments of the second reviewer are an excellent recipe for improving the paper. Furthermore, I agree with the first reviewer, that the paper could improve language, as sometimes sentences are hard to interpret. The third reviewer has only minor comments. Please correct this reviewers suggestions, but especially focus on the major concerns raised by the other two reviewers.

Furthermore, the paper is lacking a discussion paper that puts this work into perspective. Including this session is required to get the paper accepted.

---

## Author Comment (AC1) · 11 Apr 2017

**Responses to Reviewer 1:**

1) *This paper is potential and well organized, but the language proficiency still needs to be polished by Professor Sue Grimmond who is the fourth author and a native English speaker.*
**Response**: we thank the reviewer for the positive comments on our work. The revised manuscript has been proof read by Prof. Grimmond and other colleagues who are native English speakers.

2) *It will be great if authors compare their own results with the previous work.*
**Response**: A full comparison in the simulated $\Delta Q_S$ of AnOHM and OHM will be presented in an online study using the SUEWS framework. However, we added the initial comparison between AnOHM and OHM in Figure 6 and section 4.

3) *This paper is lacking a formal Discussion section. I suggest the authors develop this section.*
**Response**: The discussion part has been developed and formed in the new "Discussion and Conclusion" section.

4) *Line 14, suggest change "hampers application" to "hampers its application".*
**Response**: Changed as suggested.

5) *Line 15, change "1-dimensional" to "one-dimensional".*
**Response**: Changed as suggested.

6) *Lines 18-19, "From this albedo, Bowen ration and bulk transfer coefficient, solar radiation and wind speed are identified as being critical." I strongly recommend the authors revise this statement.*
**Response**: The statement has been rephrased as:
"The test suggests that albedo, Bowen ratio and bulk transfer coefficient, solar radiation and wind speed are identified as being critical."

7) *Line 21, change "OHM coefficients to" to "OHM coefficients".*
**Response**: Corrected as suggested.

8) *Page 2, Lines 9-10. "The volume of interest extends from the top of the roughness sub-layer to the depth in the ground where the vertical net heat conduction is zero on a daily basis (see Figure 2 in Masson et al., 2002)." The statement is contrary to its former statement.*
**Response**: We deem this statement is NOT contrary to the former statement: the former statement indicates the diurnal dynamics of heat storage in the canopy layer while the latter means the daily averaged heat storage is zero.
To better clarify our intent, this statement has been rephrased as follows:
"The volume of interest extends from the top of the roughness sub-layer to the depth in the ground where the daily average of vertical net heat conduction is zero."

9) *Page 2, Lines 11-16. It is recommended to add the references at where is appropriate, such as after "e.g. 5%", and after "the term becomes much more significant".*
**Response**: Added as suggested.

10) *Page 2, Lines 19-33, Page 3, Lines 1-17. What are the disadvantages and advantages of OHM compared with the other techniques to determine the storage heat flux? The parts of listed different techniques are verbose (Page 2, Lines 22-33).*
**Response**: Discussion on the advantages and disadvantages of OHM has been added:
"OHM features the less demanding parameterisations and more direct understanding of control of $\Delta Q_S$ by $Q^*$ compared with other approaches. Despite the shortage of OHM coefficients for the wide range facet types, OHM captures the urban SEB processes (Grimmond and Oke, 1999; Järvi et al., 2011; 2014; Karsisto et al., 2015; Roth and Oke, 1995).

11) *Page 3, Line 3. How to determine a1, a2 and a3 by observations?*
**Response**: The three coefficients are determined by least square regression between $\Delta Q_S$ and $Q^*$ observations. This has been clarified in the revised manuscript.

12) *Page 3, Lines 14-16. Suggest change the statement "Although, Gao et al. (2003; 2008) solved the 1-dimensional advection-diffusion equation of coupled heat and liquid water transport to explore the physical relation of OHM coefficients a1 and a2 to the phase lag between ΔQS and Q*," to "Although, the one-dimensional advection-diffusion equation of coupled heat and liquid water transport equation was solved by Gao et al. (2003, 2008), and the solution was used to explore the physical relation of OHM coefficients a1 and a2 to the phase lag between ΔQ S and Q* (Gao et al., 2010), ".*
**Response**: Changed as suggested.

13) *Page 3, Lines 19-26. What will be done and some results are mixed together. It is recommended that the authors revise those statements.*
**Response**: This paragraph has been rephrased as follows:
"In this paper, the solutions of the one-dimensional advection-diffusion equation of coupled heat and liquid water transport (Gao et al., 2003; 2008) are employed with the SEB (eqn 1) to investigate more fully the three OHM coefficients, the outcomes of which lead to development of the Analytical Objective Hysteresis Model (AnOHM) (section 2). Then the Monte Carlo-based Subset Simulation (Au and Beck, 2001) approach is used to undertake the sensitivity analysis of AnOHM to surface properties and hydrometeorological conditions (section 3). An offline evaluation of AnOHM's performance for five sites with different land covers (section 4) allows us to conclude that this is an alternative approach to obtain OHM coefficients. As this will allow application across a much wider range of environments and meteorological conditions, it has important implications for land surface modelling (urban and non-urban)."

14) *Page 4. t should be defined below equation (3).*
**Response**: $t$ is time and has been defined in the revised manuscript.

15) *Page 4. It is strongly recommended putting equation (7) before equation (4), as "The steady-periodic solution of equation (3) with boundary condition, Ts=A_Ts sin(wt-Y)+Ts_aver (4)".*
**Response**: Changed as recommended.

16) *Page 5. Albedo should be defined below equation (14).*
**Response**:  Defined below equation (14) as suggested.

17) *Page 5, Line 9. It is recommended adding references or state the reason that it is reasonable to assume the incoming solar radiation and air temperature follow sinusoidal forms through a day as function of the mean value for the day.*
**Response**: Reference (Sun et al., 2013) added as suggested.

18) *Page 5, equation (8). Where is the term of longwave radiation from soil surface in the longwave radiation scale (equation (8))?*
**Response**: We assume the reviewer referred to equation (10) for the outgoing longwave radiation. The longwave radiation from land surface is fully parameterised with surface temperature $T_s$ according to the Stefan–Boltzmann law and thus no extra term is introduced.

19) *Page 7, Section 2.4. I strongly recommend adding statements about the advance of the AnOHM coefficients compared with the previous OHM coefficients. Based on the abstract, to enhance physical interpretations of the OHM coefficients is one of the paper's goals.*
**Response**: Advances by AnOHM in the physical interpretation of OHM coefficients have been added in section 2.4 of the revised manuscript.

20) *Page 8, Line 20. "Stull, 1998" should be placed at "(Table 1a, based on values reported in Stull (1982))".*
**Response**: Moved as suggested.

21) *Page 9, Lines 15-17. Based on the statement "A positive (negative) S indicates an increase will lead to increase (decrease) in simulated value", an increase in albedo will increase $a_1$ and $a_3$ while decrease $a_2$ . Because Figure 2 shows that the S of $a_1$ and $a_3$ are positive and the S of $a_2$ is negative. It is strongly recommended double-checking the other statements for surface properties and the statements for hydrometerrolgical forcing parameters.*
**Response**: The original statement is correct after checking.

22) *It may be interesting to compare the S of surface properties and hydrometerrolgical forcing parameters.*
**Response**: We thank the reviewer for the suggestion. The discussion on the comparison in S between the surface properties and hydrometeorological forcing parameters has been added in section 5.

23) *It is recommended comparing the results of sensitivity analysis to previous works.*
**Response**: We thank the reviewer for the suggestion. The discussion on the comparison in *S* between the surface properties and hydrometeorological forcing parameters has been added in section 5.

24) *What does the ability of AnOHM to capture intra-annual dynamics $\Delta Q_S$ impact its simulation $\Delta Q_S$ ?*
**Response**: Such ability of AnOHM primarily improves the nocturnal magnitude of $\Delta Q_S$ (represented by $a_3$) as compared with OHM. The original $a_3$ used in OHM approach usually adopts a single value (e.g., Järvi et al. (2014)) or two values for dry and wet seasons (e.g., Ward et al. (2016)) which constrains the dynamics of $\Delta Q_S$ and in particular its nocturnal value. As the

seasonality can be well represented by solar radiation $K_\downarrow$, the inclusion of $K_\downarrow$ in the parameterisation of $a_3$ improves the nocturnal magnitude of $\Delta Q_S$.

25) *Authors should be sure to inform the reader of what may be lacking in the study as well as needs for future work.*
**Response**: The limitations of this work has been discussed in section 5 of the revised manuscript.

26) *It is recommended to add statement that how the current work actually advances science.*
**Response**: The perspectives of this work has been discussed in section 5 of the revised manuscript.

**References:**
Au, S. K. and Beck, J. L.: Estimation of small failure probabilities in high dimensions by subset simulation, Probabilistic Engineering Mechanics, 16(4), 263–277, doi:10.1016/S0266-8920(01)00019-4, 2001.

Gao, Z., Fan, X. G. and Bian, L. G.: An analytical solution to one-dimensional thermal conduction-convection in soil, Soil Science, 168(2), 99–107, doi:10.1097/01.ss.0000055305.23789.be, 2003.

Gao, Z., Lenschow, D. H., Horton, R., Zhou, M., Wang, L. and Wen, J.: Comparison of two soil temperature algorithms for a bare ground site on the Loess Plateau in China, J Geophys Res-Oc Atm, 113(D18), D18105, 2008.

Grimmond, C. S. B. and Oke, T. R.: Heat storage in urban areas: local-scale observations and evaluation of a simple model, J Appl Meteorol, 38(7), 922–940, doi:10.1175/1520-0450(1999)038<0922:HSIUAL>2.0.CO;2, 1999.

Järvi, L., Grimmond, C. S. B. and Christen, A.: The Surface Urban Energy and Water Balance Scheme (SUEWS): Evaluation in Los Angeles and Vancouver, J Hydrol, 411(3-4), 219–237, doi:10.1016/j.jhydrol.2011.10.001, 2011.

Järvi, L., Grimmond, C. S. B., Taka, M., Nordbo, A., Setälä, H. and Strachan, I. B.: Development of the Surface Urban Energy and Water Balance Scheme (SUEWS) for cold climate cities, Geoscientific Model Development, 7(4), 1691–1711, doi:10.5194/gmd-7-1691-2014, 2014.

Karsisto, P., Fortelius, C., Demuzere, M., Grimmond, C. S. B., Oleson, K. W., Kouznetsov, R., Masson, V. and Järvi, L.: Seasonal surface urban energy balance and wintertime stability simulated using three land-surface models in the high-latitude city Helsinki, Q.J.R. Meteorol. Soc., 142(694), 401–417, doi:10.1002/qj.2659, 2015.

Roth, M. and Oke, T. R.: Relative Efficiencies of Turbulent Transfer of Heat, Mass, and Momentum over a Patchy Urban Surface, J. Atmos. Sci., 52(11), 1863–1874, doi:10.1175/1520-0469(1995)052<1863:reotto>2.0.co;2, 1995.

Sun, T., Wang, Z.-H. and Ni, G.-H.: Revisiting the hysteresis effect in surface energy budgets, Geophys. Res. Lett., 40(9), 1741–1747, doi:10.1002/grl.50385, 2013.

Ward, H. C., Kotthaus, S., Järvi, L. and Grimmond, C. S. B.: Surface Urban Energy and Water Balance Scheme (SUEWS): Development and evaluation at two UK sites, Urban Climate, 18, 1–32, doi:10.1016/j.uclim.2016.05.001, 2016.

---

## Author Comment (AC2) · 11 Apr 2017

**Responses to Reviewer 2:**

**We appreciate the generally positive comments and constructive suggestions from the reviewer. Our detailed responses are given after each comment (*italics)* below.**

1) *My main concern with the paper is the readability of the paper. In general the paper lacks a justification of the utilized methodologies (especially the parameter estimation, LOESS method etc) and complete description of these method. In terms of style, the paper reads a bit as a flood on information on equations and parameters, but a real interpretation of the results is missing. Overall as a reader I get too much a feeling that the whole paper provides a black box approach.*

**Response**: We have improved the overall readability of this paper in the following aspects:
   a. The justification of utilized methodologies, including the parameter estimation, LOESS method and determination of OHM coefficients, has been added;
   b. A discussion with more interpretations of the results has been added.

2) *Interpretation: The followed approach provides new values and uncertainties in the parameter values of the OHM model. However, the paper does not reach a level beyond these parameter values. I think the reader expects more interpretation on the various parameter values and how much it would change the surface energy balance as a whole by the new information at hand. Moreover, the bias and RMSE are still quite high for some of the presented sites. I miss an outlook on how the authors will further address this, or any hypothesis behind these biases.*

**Response**: More physical interpretations of the new formulations of AnOHM coefficients in section 2.4. Also, the outlook for potential use of AnOHM has been added in section 5.

3) *The paper is missing a discussion section. The authors can be more critical towards their results, the influence of certain assumptions made in the analysis on the results (e.g. assuming e=ea=es =0.85). Moreover AnOHM should outperform the original OHM, but this is not shown.*

**Response**: The limitations of the AnOHM framework, including the assumption $\varepsilon_a \approx \varepsilon_s \approx \varepsilon$, have been added in section 5.

For the performance of AnOHM against the original OHM, we have partially addressed this concern in section 4 by demonstrating the ability of AnOHM in capturing the seasonality of the coefficients (see Anandakumar (1999) for observational evidence). Furthermore, in the revised manuscript, we have added the comparison in these coefficients by different modelling and observational regression approaches as Appendix B, indicating AnOHM generally follows the results by observation regression, whereas the typical coefficient values adopted by OHM do not (Figure R1).

[Figure]

**Figure R1** Comparison in OHM coefficients (left, central and right columns for $a_1$, $a_2$ and $a_3$, respectively) between different modelling approaches and observation regression at five sites: UK-Ldn (a, b, c), US-Wlr (d, e, f), CA-NS5 (g, h, i), US-SRM (j, k, l) and US-SO4 (m, n, o). The blue dots denote the pairing values between AnOHM and observation regression. The orange lines represent the reference value used in OHM simulations for land covers with grass and tree (Grimmond and Oke, 1999), whereas the green lines shows median values derived from results by observation regression at corresponding sites.

4) *In equations (10) and (26) the upwelling component e_s\*L_down is missing. How does this missing component affect the paper's results and parameter sensitivities, especially to es?*
**Response**: We thank the reviewer for raising the valuable question about upwelling longwave radiation parameterisation.

In the formulation of outgoing longwave radiation $L_\uparrow$, a simplified form (i.e., $\varepsilon_s \sigma T_s^4$) is used for AnOHM as eqn 10 by ignoring the reflected part of $L_\downarrow$ (i.e., $(1 - \varepsilon_s)L_\downarrow$). The rationale for such simplification is explained that given $\varepsilon_s$ is usually larger than 0.9, $(1 - \varepsilon_s)L_\downarrow$ contributes a relatively small portion to the total longwave component (Oke, 1987) and omission of this part is well accepted in the parameterization of upwelling longwave radiation for land surface modeling across various land covers (Bateni and Entekhabi, 2012; Lee et al., 2011).

Using the parameterisation of incoming longwave radiation in the AnOHM framework (i.e., $L_\downarrow = \varepsilon_a \sigma T_a^4 \approx \varepsilon_s \sigma T_a^4$), we conduct a sensitivity analysis of the ratio between the ignored part (i.e., $(1 - \varepsilon_s)L_\downarrow$) and total upwelling longwave radiation (i.e., $\varepsilon_s \sigma T_s^4 + (1 - \varepsilon_s)L_\downarrow$) at a constant air temperature of 20 °C and find this ratio is generally less than 5% given $\varepsilon_s$ ranges between 0.90 and 0.99 (Figure R2).

[Figure]

**Figure R2** Ratio between reflected part (i.e., $(1 - \varepsilon_s)L_\downarrow$) and total upwelling longwave radiation (i.e., $\varepsilon_s \sigma T_s^4 + (1 - \varepsilon_s)L_\downarrow$) at a constant air temperature of 20 °C.

Moreover, if $(1 - \varepsilon_s)L_\downarrow$ is included in the net longwave radiation, the induced effect can be incorporated into a modified sky emissivity $\varepsilon_a' = \varepsilon_s \varepsilon_a$ as follows:
$$
\begin{aligned}
L_{net} &= L_\downarrow - L_\uparrow \\
&= L_\downarrow - (\varepsilon_s \sigma T_s^4 + (1 - \varepsilon_s)L_\downarrow) \\
&= \varepsilon_s L_\downarrow - \varepsilon_s \sigma T_s^4 \\
&= \varepsilon_s \varepsilon_a \sigma T_a^4 - \varepsilon_s \sigma T_s^4 \\
&= \varepsilon_a' \sigma T_a^4 - \varepsilon_s \sigma T_s^4
\end{aligned}
$$
Then by assuming $\varepsilon \approx \varepsilon_a' \approx \varepsilon_s$, the derivation following equation 18 still holds. Also, the sensitivity analysis suggests that the derived coefficients are insensitive to $\varepsilon$ (cf. $S$ for $\varepsilon$ in Figure 2 of the manuscript).
As such, we deem the omission of $(1 - \varepsilon_s)L_\downarrow$ will not qualitatively change the results of this work.
The above discussion has been added in the revised manuscript.

5) *Equation 21, first line: I have the impression the 4's should be removed (or the last two terms should be replaced by 4\*sigma\*eT^3(Ts-Ta)).*
**Response**: We thank the reviewer for pointing out this typo. The 4's have been removed in the revised manuscript.

6) *P11, ln 15: I find the hit rate not a good metric to evaluate this model, at least not if presented as the only metric. In terms of contingency tables, the hit rate should always be*

*presented together with the false-alarm rate, and preferably with an critical success index or a threat score.*

**Response**: As the hit rate may bring up confusion to the readers, we have removed this metric but kept the other two (i.e., mean bias and RMSE) in the revised manuscript.

**References:**

Anandakumar, K.: A study on the partition of net radiation into heat fluxes on a dry asphalt surface, Atmos. Environ., 33(24–25), 3911–3918, doi:10.1016/S1352-2310(99)00133-8, 1999.

Bateni, S. M. and Entekhabi, D.: Relative efficiency of land surface energy balance components, Water Resour. Res., 48(4), W04510, doi:10.1029/2011WR011357, 2012.

Grimmond, C. S. B. and Oke, T. R.: Heat storage in urban areas: local-scale observations and evaluation of a simple model, J Appl Meteorol, 38(7), 922–940, doi:10.1175/1520-0450(1999)038<0922:HSIUAL>2.0.CO;2, 1999.

Lee, X., Goulden, M. L., Hollinger, D. Y., Barr, A., Black, T. A., Bohrer, G., Bracho, R., Drake, B., Goldstein, A., Gu, L., Katul, G. G., Kolb, T., Law, B. E., Margolis, H., Meyers, T., Monson, R., Munger, W., Oren, R., U, K. T. P., Richardson, A. D., Schmid, H. P., Staebler, R., Wofsy, S. and Zhao, L.: Observed increase in local cooling effect of deforestation at higher latitudes, Nature, 479(7373), 384–387, doi:10.1038/nature10588, 2011.

Oke, T. R.: Boundary Layer Climates, Taylor & Francis, Abingdon, UK. 1987.

---

## Author Comment (AC3) · 11 Apr 2017

**Responses to Reviewer 3:**

**We appreciate the generally positive comments and constructive suggestions from the reviewer. Our detailed responses are given after each comment (*italics)* below.**

1) *Page 1, line 14, it is recommended describing OHM limitation more clearly.*
**Response**: Due to the limitation in space of the abstract, we have elaborated the limitations of OHM in coefficient availability in the introduction of the revised manuscript.

2) *Please check Eq.27 in page 7. Does it maybe $a_{3F} = -a_1 \frac{f_T}{f}(1-\alpha)\overline{K}_\downarrow - a_1 Q_F$?*

   *Based on Eq.22, $\Delta Q_S = a_1(Q^* + Q_F) + a_2 \frac{\partial(Q^*+Q_F)}{\partial t} + a_{3F}$ when $Q_F$ is included. With the assumption that $Q_F$ is diurnal invariant,*

$$\Delta Q_S = a_1(Q^* + Q_F) + a_2 \frac{\partial(Q^* + Q_F)}{\partial t} + a_{3F}$$
$$= a_1 Q^* + a_2 \frac{\partial Q^*}{\partial t} + a_1 Q_F + a_{3F}$$

   *so $a_3 = a_1 Q_F + a_{3F}$o, and $a_{3F} = a_3 - a_1 Q_F = -a_1 \frac{f_T}{f}(1-\alpha)\overline{K}_\downarrow - a_1 Q_F$.*

**Response**: We thank the reviewer for providing another derivation of $a_{3F}$. However, the reviewer's derivation is NOT within the framework of AnOHM/OHM, whose aim is to establish the relationship between the heat storage $\Delta Q_S$ and net radiation $Q^*$, rather than the sum of $Q^*$ and anthropogenic heat $Q_F$ (i.e., $Q^* + Q_F$). In fact, by replacing equation 14 with equation 26 and following the steps in section 2.2, one can finally obtain equation 27.

3) *In page 10, a greater in incoming solar radiation ($K_\downarrow$) will lead to smaller $\Delta Q_S$, why? In general, net radiation mostly depends on $K_\downarrow$, and the larger $K_\downarrow$, the larger net radiation which will lead to larger $\Delta Q_S$.*
**Response**: We note that we do NOT mean the larger $K_\downarrow$ will lead to "the smaller $\Delta Q_S$" (as interpreted by the reviewer) but "a smaller portion of (the solar energy will be partitioned) to $\Delta Q_S$". In other words, it is NOT the smaller *absolute magnitude* of $\Delta Q_S$ but the smaller *partitioning fraction* of $\Delta Q_S$ that will be resulted in given a larger $K_\downarrow$.

4) *In Figure 5, the blue solid line (URB) is large differently from other lines in (a) and (c). Based on Figure 5a, the $\Delta Q_S$ can be up to 70% of net radiation, it's too large to believe. In addition, there's also large difference between simulation and observation in Figure 5a, 5b. Please explain them.*
**Response**: The two concerns of the reviewer are addressed as follows:
   a. *Too large ratio (e.g., 0.7) between $\Delta Q_S$ and $Q^*$ to believe*:
   Figure 5a shows the coefficient $a_1$, which essentially is NOT the ratio between $\Delta Q_S$ and $Q^*$ but rather characterize such ratio. That being said, high values of $a_1$ (>0.7) have been reported for urban environments in literature (e.g., Doll et al. (1985), Grimmond and Oke (1999)). As such, the values of $a_1$ reported here are not as surprising as the reviewer claimed to be.
   b. *Large difference between simulation and observation in Figure 5a and 5b*:

First, Figure 5 is meant to demonstrate the seasonality in the AnOHM/OHM coefficients rather than the comparison in such coefficients between observations and predictions as they are based on different sites/land covers. In other words, the key message delivered by Figure 5 is that the AnOHM/OHM coefficients vary between seasons within a year and their seasonality should thus be considered in conducting OHM simulations.

Besides, the large differences in $a_1$ (Figure 5a) and $a_2$ (Figure 5b) between Anandakumar (1999) and the other sites inherently imply the distinct impacts of different land covers/surface status on energy partitioning, which is widely observed and well reported in literature (e.g., Li et al. (2015), Bateni and Entekhabi (2012), Teuling et al. (2010)). Also, we note that the negative values of $a_2$ observed Anandakumar (1999) (squares in Figure 5b) can be explained by the phase difference between $\Delta Q_S$ and $Q^*$ (see equation 24) and a more detailed discussion on this phase difference is referred to Sun et al. (2013).

**References:**

Anandakumar, K.: A study on the partition of net radiation into heat fluxes on a dry asphalt surface, Atmos. Environ., 33(24–25), 3911–3918, doi:10.1016/S1352-2310(99)00133-8, 1999.

Bateni, S. M. and Entekhabi, D.: Relative efficiency of land surface energy balance components, Water Resour. Res., 48(4), W04510, doi:10.1029/2011WR011357, 2012.

Doll, D., Ching, J. and Kaneshiro, J.: Parameterization of subsurface heating for soil and concrete using net radiation data, Boundary-Layer Meteorol., 32(4), 351–372, doi:10.1007/BF00122000, 1985.

Grimmond, C. S. B. and Oke, T. R.: Heat storage in urban areas: local-scale observations and evaluation of a simple model, J Appl Meteorol, 38(7), 922–940, doi:10.1175/1520-0450(1999)038<0922:HSIUAL>2.0.CO;2, 1999.

Li, D., Sun, T., Liu, M., Yang, L., Wang, L. and Gao, Z.: Contrasting responses of urban and rural surface energy budgets to heat waves explain synergies between urban heat islands and heat waves, Environ Res Lett, 10(5), 054009, doi:10.1088/1748-9326/10/5/054009, 2015.

Sun, T., Wang, Z.-H. and Ni, G.-H.: Revisiting the hysteresis effect in surface energy budgets, Geophys. Res. Lett., 40(9), 1741–1747, doi:10.1002/grl.50385, 2013.

Teuling, A. J., Seneviratne, S. I., Stöckli, R., Reichstein, M., Moors, E., Ciais, P., Luyssaert, S., van den Hurk, B., Ammann, C., Bernhofer, C., Dellwik, E., Gianelle, D., Gielen, B., Grünwald, T., Klumpp, K., Montagnani, L., Moureaux, C., Sottocornola, M. and Wohlfahrt, G.: Contrasting response of European forest and grassland energy exchange to heatwaves, Nature Geosci, 3(10), 722–727, doi:10.1038/ngeo950, 2010.

---

## Author Comment (AC4) · 11 Apr 2017

Dear Dr. van Heerwaarden,

Your constructive suggestions for the revision are much appreciated. Now we have addressed the concerns of the three reviewers and have added more discussion per request in the revised manuscript.

We thank you again for your time and we look forward to hearing from you.

Respectfully,

Ting Sun on behalf of all co-authors

---

## Author Comment (AC5) · 11 Apr 2017

Please see the attached document with changes highlighted.

Please also note the supplement to this comment:
http://www.geosci-model-dev-discuss.net/gmd-2016-300/gmd-2016-300-AC5-supplement.pdf

---

## Author Response (AR2)

**Responses to Editor:**

**Thank you for your constructive suggestions. Our responses to each comment (*italics)* are given below .**

1) *Please follow the reviewer's advice to motivate your study better in the introduction. I do not think the research question is essential, as it is a model description paper, but nonetheless the purpose of your improved model has to be really clear.*

**Response**: We have added additional text. The rationale for this study is to extend the applicability of OHM in modelling storage heat flux $\Delta Q_S$ for areas (land uses and land covers) where measurements are not available and to cover a wider range of seasonal/meteorological conditions. We have elaborated on this in the manuscript:
   a. The importance of $\Delta Q_S$: page 2 lines 5–17.
   b. The advantages and limitations of OHM: page 3 lines 1–16.
   c. The lack in the physical interpretations of OHM coefficients: page 3 lines 16–18.

2) *I would like you to take into account the suggestions of the reviewer concerning the equations. Please make clear where equations are taken from textbooks and where they are your own work.*

**Response**: We have added additional references to make this clear. Notably, references (Gao et al., 2003; 2010) for the solution (eqns 4 and 5) have been added in the revised manuscript.

3) *The reviewer's comments on the outgoing longwave radiation require some additional work, please provide physical justification for ignoring the term.*

**Response**:  First of all, we acknowledge that inclusion of the "re-emitted downwelling longwave radiation" improves the physical rigour of the parameterisation of outgoing longwave radiation $L_\uparrow$. We now include this as part 2 in eqn 10 of the revised manuscript. However, as the omission of this term is well accepted in modelling the outgoing longwave radiation (Bateni and Entekhabi, 2012; Lee et al., 2011; Stensrud, 2007), and its omission greatly enhances the simplicity of the AnOHM formulations, the simplified form of $L_\uparrow$ (i.e., $\varepsilon_s \sigma T_s^4$) is still used in this study. The rationale for using the simplified form is presented in Appendix A.

We do not, though, agree with the following comment by the reviewer that
*"the longwave radiation is not reflected (solely possible for shortwave radiation), but re-emitted."*

We recognise that the concept of reflectivity is only valid for the case of a single wavelength; however, in practise it is referred to in this way for fairly wide wavebands (e.g., such integral reflectivity is referred to as *albedo* for shortwave radiation). However, longwave radiation can be reflected given we are not concerned with ideal blackbody surfaces. The related physics is discussed in section 3 of chapter 1 by Oke (1987).

4) *It would also be great if you can explain in your introduction in a single sentence what exactly an OHM is and on which physical principles it is based.*

**Response**: This explanation has been provided on page 2 lines 29–30 of the introduction: OHM is a model to estimate the storage heat flux $\Delta Q_S$ based on a hysteresis relation between $\Delta Q_S$ and net radiation $Q^*$.

5) *It would be useful to have again a native speaker check your paper.*

**Response**: This has been done.

**References:**

Bateni, S. M. and Entekhabi, D.: Relative efficiency of land surface energy balance components, Water Resour. Res., 48(4), W04510, doi:10.1029/2011WR011357, 2012.

Gao, Z., Fan, X. G. and Bian, L. G.: An analytical solution to one-dimensional thermal conduction-convection in soil, Soil Science, 168(2), 99–107, doi:10.1097/01.ss.0000055305.23789.be, 2003.

Gao, Z., Horton, R. and Liu, H. P.: Impact of wave phase difference between soil surface heat flux and soil surface temperature on soil surface energy balance closure, J. Geophys. Res., 115(D16), D16112, doi:10.1029/2009JD013278, 2010.

Lee, X., Goulden, M. L., Hollinger, D. Y., Barr, A., Black, T. A., Bohrer, G., Bracho, R., Drake, B., Goldstein, A., Gu, L., Katul, G. G., Kolb, T., Law, B. E., Margolis, H., Meyers, T., Monson, R., Munger, W., Oren, R., U, K. T. P., Richardson, A. D., Schmid, H. P., Staebler, R., Wofsy, S. and Zhao, L.: Observed increase in local cooling effect of deforestation at higher latitudes, Nature, 479(7373), 384–387, doi:10.1038/nature10588, 2011.

Oke, T. R.: Boundary Layer Climates, Taylor & Francis, Abingdon, UK. 1987.

Stensrud, D. J.: Parameterization schemes: keys to understanding numerical weather prediction models, Cambridge University Press, Cambridge. 2007.

**Responses to Reviewer 2:**

**We appreciate the comments and constructive suggestions from the reviewer. Our detailed responses are given after each comment (*italics*) below.**

***Major comment:***
1) *The authors have revised the manuscript. However, I have the feeling the revised version still contains the main deficiencies that I have mentioned in my previous review. The justification for ignoring the re-emitted downwelling longwave radiation in the upwelling longwave radiation has been addressed, but still ignoring this term is physically wrong. In addition, the readability of the paper has not improved in my view; it is still a long list of equations that are sometimes poorly connected. For example section 3.1 could have literally been copied from a mathematics book, but the link with the AnOHM parameter estimation is missing. I suggest the authors improve the description of all steps that lead to the analytical solutions provided. Moreover the paper lacks a clear research question and some justification for the need to do this research, i.e. what can now be done with the AnOHM that was not possible before with the original OHM, and why an analytical approach is the most feasible to answer the research question.*

**Response**: We would like to thank the reviewer for his/her constructive comments and critique, which provides us another opportunity to elaborate our thinking.

The reviewer's comments can be summarised as:
1. Ignoring the re-emitted downwelling longwave radiation is physically wrong.
2. The readability of this manuscript needs to be improved.
3. The motivation of this study is unclear.

Our responses to these points are:
1. We recognise that inclusion of the "re-emitted downwelling longwave radiation" improves the physical rigour of the parameterisation of outgoing longwave radiation $L_\uparrow$. This now is included as part 2 in eqn 10 of the revised manuscript.
   However, as the omission of this term is very well accepted in modelling the outgoing longwave radiation (Bateni and Entekhabi, 2012; Lee et al., 2011; Stensrud, 2007), and its omission does greatly enhance the simplicity of AnOHM formulations, the simplified form of $L_\uparrow$ (i.e., $\varepsilon_s \sigma T_s^4$) is still used in this study. The rationale for using the simplified form is presented in Appendix A.
2. We have checked the steps thoroughly and added explanations where we believe they are necessary.
3. The rationale for this study is to extend the applicability of OHM in modelling storage heat flux $\Delta Q_S$ for land covers and time periods where measurements are not available. We have elaborated on this in the manuscript:
   a. The importance of $\Delta Q_S$: page 2 lines 5–17.
   b. The advantages and limitations of OHM: page 3 lines 1–16.
   c. The lack in the physical interpretations of OHM coefficients: page 3 lines 16–18.

***Other comments:***
2) *Eq. 4 and 5 should be correctly referenced, since this is not a solution of your own course.*

**Response**: References (Gao et al., 2003; 2010) for the solution (eqns 4 and 5) have been added in the revised manuscript.

3) *P13, ln 3: the longwave radiation is not reflected (solely possible for shortwave radiation), but re-emitted.*

**Response**: we recognise that the concept of reflectivity is only valid for the case of a single wavelength; however, in practise it is often referred to in this way for fairly wide wavebands (e.g., such integral reflectivity is referred to as *albedo*). Longwave radiation can be reflected given we are not concerned with ideal blackbody surfaces. The related physics is discussed (for example) in section 3 of chapter 1 by Oke (1987).

[revised manuscript text omitted]

**2.2 Parameterization of Net All-wave Radiation $Q^*$ for a Land Surface**

Given the parameterizations of incoming longwave radiation $L_\downarrow$, outgoing longwave radiation $L_\uparrow$, sensible heat flux $Q_H$, latent heat flux $Q_E$, and storage heat flux $\Delta Q_S$ as follows:

$$L_\downarrow = \varepsilon_a \sigma T_a^4, \tag{9}$$

$$L_\uparrow = \underbrace{\varepsilon_s \sigma T_S^4}_{1} + \underbrace{(1 - \varepsilon_S)L_\downarrow}_{2}, \tag{10}$$

$$Q_H = C_h U(T_S - T_a), \tag{11}$$

$$Q_E = Q_H/\beta, \tag{12}$$

$$\Delta Q_S = G_0, \tag{13}$$

the boundary condition imposed by the SEB relation can be rewritten as:

$$(1 - \alpha)K_\downarrow + \varepsilon_a \sigma T_a^4 - \varepsilon_s \sigma T_S^4 = C_h U(1 + \beta^{-1})(T_S - T_a) + G_0 \tag{14}$$

where the turbulent fluxes $Q_H$ and $Q_E$ are parameterized as functions of temperature gradient $T_S - T_a$ with albedo $\alpha$, bulk transfer coefficient $C_h$, wind speed $U$ and Bowen ratio ($\beta = Q_H/Q_E$). Theoretically, the second part of eqn 10 (i.e. $(1 - \varepsilon_s)L_\downarrow$) should be accounted for in the estimation of $L_\uparrow$ (Oke, 1987), however, given it is usually less than $\sim$5% of the first part of the equation (see full discussion in Appendix A) for most land covers (Oke, 1987), here it is omitted from consideration and in the development of AnOHM.

[revised manuscript text omitted]